# Regulation of multiple signaling pathways promotes the consistent expansion of human pancreatic progenitors in defined conditions

Luka Jarc[1,2†], Manuj Bandral[1,2†], Elisa Zanfrini[1,2], Mathias Lesche[3,4], Vida Kufrin[1], Raquel Sendra[1], Daniela Pezzolla[2,5], Ioannis Giannios[1,2], Shahryar Khattak[6‡], Katrin Neumann[6], Barbara Ludwig[1,2,5,7], Anthony Gavalas[1,2]*

[1]Paul Langerhans Institute Dresden (PLID) of Helmholtz Center Munich at the University Clinic Carl Gustav Carus of TU Dresden, Helmholtz Zentrum München, German Research Center for Environmental Health, Neuherberg, Germany; [2]German Centre for Diabetes Research (DZD), Munich, Germany; [3]Dresden Concept Genome Centre (DcGC), TU Dresden, Dresden, Germany; [4]Center for Molecular and Cellular Bioengineering (CMCB) Technology Platform, TU Dresden, Dresden, Germany; [5]Center for Regenerative Therapies Dresden (CRTD), Faculty of Medicine, TU Dresden, Dresden, Germany; [6]Stem Cell Engineering Facility, (SCEF), CRTD, Faculty of Medicine, TU Dresden, Dresden, Germany; [7]Department of Medicine III, University Hospital Carl Gustav Carus and Faculty of Medicine, TU Dresden, Dresden, Germany

*For correspondence:
anthony.gavalas@tu-dresden.de

[†]These authors contributed equally to this work

Present address: [‡]KAUST Smart-Health Initiative (KSHI) and Biological & Environmental Science and Engineering (BESE) Division, King Abdullah University of Science & Technology (KAUST), Thuwal, Saudi Arabia

**Competing interest:** The authors declare that no competing interests exist.

**Abstract** The unlimited expansion of human progenitor cells in vitro could unlock many prospects for regenerative medicine. However, it remains an important challenge as it requires the decoupling of the mechanisms supporting progenitor self-renewal and expansion from those mechanisms promoting their differentiation. This study focuses on the expansion of human pluripotent stem (hPS) cell-derived pancreatic progenitors (PP) to advance novel therapies for diabetes. We obtained mechanistic insights into PP expansion requirements and identified conditions for the robust and unlimited expansion of hPS cell-derived PP cells under GMP-compliant conditions through a hypothesis-driven iterative approach. We show that the combined stimulation of specific mitogenic pathways, suppression of retinoic acid signaling, and inhibition of selected branches of the TGFβ and Wnt signaling pathways are necessary for the effective decoupling of PP proliferation from differentiation. This enabled the reproducible, 2000-fold, over 10 passages and 40–45 d, expansion of PDX1[+]/SOX9[+]/NKX6-1[+] PP cells. Transcriptome analyses confirmed the stabilization of PP identity and the effective suppression of differentiation. Using these conditions, PDX1[+]/SOX9[+]/NKX6-1[+] PP cells, derived from different, both XY and XX, hPS cell lines, were enriched to nearly 90% homogeneity and expanded with very similar kinetics and efficiency. Furthermore, non-expanded and expanded PP cells, from different hPS cell lines, were differentiated in microwells into homogeneous islet-like clusters (SC-islets) with very similar efficiency. These clusters contained abundant β-cells of comparable functionality as assessed by glucose-stimulated insulin secretion assays. These findings established the signaling requirements to decouple PP proliferation from differentiation and allowed the consistent expansion of hPS cell-derived PP cells. They will enable the establishment of large banks of GMP-produced PP cells derived from diverse hPS cell lines. This approach will streamline SC-islet production for further development of the differentiation process, diabetes research, personalized medicine, and cell therapies.

### eLife assessment

This **important** study describes a method to decouple the mechanisms supporting pancreatic progenitor self-renewal and expansion from feed-forward mechanisms promoting their differentiation allowing in vitro expansion of hPSC-derived pancreatic progenitors. The strength of evidence is **convincing** in that the authors use appropriate and validated methodology in line with current state-of-the-art. The work will be of interest to the field of β-cell replacement therapy in diabetes.

## Introduction

Diabetes is a global epidemic affecting almost 10% of the world's population. Its two main forms result from the immune destruction (type 1) or malfunction (type 2) of insulin-producing β-cells residing in the pancreatic islets of Langerhans. Severe cases of diabetes often necessitate whole pancreas transplantation or pancreatic islet transplantation to restore metabolic control and insulin independence. However, both approaches face critical limitations due to the scarcity of tissue donors and the requirement for lifelong immunosuppression that hinders the already limited β-cell self-renewal and is associated with severe side effects and morbidity (*Nir et al., 2007*).

The remarkable progress over the last decade in the differentiation of human pluripotent stem (hPS) cells into pancreatic islet cells (SC-islet cells) suggests that this approach could provide an unlimited source of β-cells for transplantations and personalized medicine (*Amin et al., 2018*; *Balboa et al., 2022*; *Du et al., 2022*; *Hogrebe et al., 2020*; *Millman et al., 2016*; *Ramzy et al., 2021*; *Shapiro et al., 2021*). The underlying strategy recapitulates in vitro the stepwise differentiation of epiblast cells initially to definitive endoderm and then to pancreatic progenitor-containing (PP) cells, pancreatic endocrine progenitors, and, finally, to pancreatic islet cells (*Kroon et al., 2008*; *Pagliuca et al., 2014*; *Rezania et al., 2014*; *Russ et al., 2015*; *Zeng et al., 2016*; *Zhu et al., 2016a*). Clinical trials employing either GMP-grade PP cells in a non-immunoprotective macroencapsulation device (*Shapiro et al., 2021*) (NCT03163511) or GMP-grade SC-islet cells in an immunoprotective macroencapsulation device (NCT02239354) suggested that diabetes cell therapies can become a reality. Obstacles to be overcome include the limited maturation of the resulting β-cells, incomplete conversion of hPS cells into endocrine cells, and the large number of SC-islet cells required for a single transplantation. Elucidating the mechanisms that maintain the self-renewal of pure PP cells, while inhibiting their differentiation, would allow the establishment of expandable populations of PP cells and address the need for large numbers of pancreatic endocrine cells.

During development, the intersection of several signals induces the pancreatic anlage at the posterior foregut. Repression of posteriorly derived Wnt signaling is initially essential to define the foregut region (*McLin et al., 2007*). There, the pancreas and liver develop from a common progenitor (*Cerdá-Esteban et al., 2017*; *Deutsch et al., 2001*). Bipotent progenitors, located distant to the cardiac mesoderm, which secretes the pro-hepatic signals FGF10 and BMPs, remain competent to adopt pancreatic fate (*Jung et al., 1999*; *Rossi et al., 2001*). The combination of RA signaling, derived from the somitic mesoderm (*Martín et al., 2005*; *Molotkov et al., 2005*), and endothelial signals, derived from the dorsal aorta (*Lammert et al., 2001*), induces the formation of PPs. A complex, stage-specific expression of several transcription factors (TFs) guides pancreatic cell development (*Duvall et al., 2022*). The epithelial PPs are characterized by the combined expression of several TFs, most notably Pdx1, Nkx6-1, and Sox9, which are essential for progenitor self-renewal and subsequent endocrine differentiation. Pdx1 expression is induced at the boundary between the Sox2-expressing anterior endoderm and the Cdx2-expressing posterior endoderm, and it is necessary to maintain the pancreatic identity (*Sherwood et al., 2009*). Its function is reinforced by Sox9, which cooperates with Pdx1 to repress Cdx2 expression in the pancreatic anlage (*Shih et al., 2015*). Pdx1 (*Jonsson et al., 1994*; *Offield et al., 1996*) and Sox9 (*Seymour et al., 2007*) loss-of-function experiments resulted in pancreatic agenesis, establishing their key role in maintaining pancreatic identity. Sox9 is subsequently essential for the induction of Neurog3, the TF, which is necessary and sufficient for the induction of the pancreatic endocrine lineage (*Gradwohl et al., 2000*; *Grapin-Botton et al., 2001*). Nkx6-1 is another key contributor to the maintenance and self-renewal of PPs (*Sander et al., 2000*). It is also required for β-cell specification since ectopic Nkx6-1 expression directed endocrine precursors into β-cells, whereas β-cell-specific ablation of Nkx6-1 diverted these cells into the other endocrine lineages (*Schaffer et al., 2013*). Given the importance of these three TFs, the identification

of conditions to stabilize PDX1$^+$/SOX9$^+$/NKX6-1$^+$ cells will set the stage for unlimited PP cell expansion and their efficient differentiation into SC-islets.

PDX1$^+$/SOX9$^+$/NKX6-1$^+$ PP cells undergo self-renewal in vivo for a limited amount of time. During development, feed-forward loops that steer cells toward differentiation operate in parallel with maintenance and self-renewal mechanisms. To achieve unlimited expansion of PP cells in vitro, it will be necessary to disentangle differentiation signals from proliferation and maintenance signals. Several pathways have been implicated in these processes. Notch signaling mediates progenitor self-renewal as well as lineage segregation (*Afelik et al., 2012*; *Murtaugh et al., 2003*; *Shih et al., 2012*), and its downregulation is necessary for the endocrine lineage specification (*Apelqvist et al., 1999*; *Jensen et al., 2000*). Notch signaling is itself regulated by the extracellular signal sphingosine-1-phosphate (S1p) (*Serafimidis et al., 2017*). Canonical Wnt signaling has also been implicated in PP maintenance (*Afelik et al., 2015*). Low levels of endogenous RA signaling are involved in subsequent differentiation steps of PP cells (*Kobayashi et al., 2002*; *Lorberbaum et al., 2020*; *Tulachan et al., 2003*; *Vinckier et al., 2020*). Finally, expression of TGFβ ligands and receptors is widespread during pancreas development (*Crisera et al., 1999*; *Tulachan et al., 2007*), and it has been suggested that the TGFβ pathway activity regulates the pancreatic lineage allocation (*Miralles et al., 1998*; *Sanvito et al., 1994*). TEAD, and its coactivator YAP, acts as integrators of extracellular signals to activate key pancreatic signaling mediators and TFs, which promote the expansion of PPs and their competence to differentiate into endocrine cells (*Cebola et al., 2015*; *Serafimidis et al., 2017*). Cell confinement acts as a mechanic signal to downregulate YAP and trigger endocrine differentiation (*Mamidi et al., 2018*), and this finding has been applied to promote the differentiation of hPS cell-derived PPs into endocrine cells (*Rosado-Olivieri et al., 2019*).

Culture conditions to expand hPS cell-derived PDX1$^+$/SOX9$^+$ cells have been previously reported, but these methods relied either on feeder layers or limiting 3D conditions in an agarose hydrogel matrix and did not maintain the expression of the crucial TF NKX6-1 (*Konagaya and Iwata, 2019*; *Trott et al., 2017*). A recent study claimed the expansion of a hPS cell-derived cell population, containing PDX1$^+$/NKX6-1$^+$ cells, on fibronectin and in a defined medium using SB431542, a broad (ALK4/5/7) TGFβ inhibitor (*Inman et al., 2002*; *Nakamura et al., 2022*). Unfortunately, the reproducibility of the expansion was not documented and the reported percentage of PDX1$^+$/NKX6-1$^+$ PP cells in the expanded cells varied widely among the three different hPS cell lines used, from 65% to 35% and 20%. Furthermore, RNA-Seq data showed that *NKX6-1* expression actually decreased during expansion in two out of the three samples presented (E-MTAB-9992), suggesting that NKX6-1 expression was not reliably maintained under those culture conditions. In another study, PP cells, generated through genetic reprogramming of human fibroblasts into endoderm progenitor cells, could be expanded in a chemically defined medium containing epithelial growth factor (EGF), basic fibroblast growth factor (bFGF), and SB431542. However, the percentage of PDX1$^+$/NKX6-1$^+$ cells was less than 20% (*Zhu et al., 2016b*). The authors then identified a BET bromodomain inhibitor that could mediate the unlimited expansion of PDX1$^+$/NKX6-1$^+$ PP cells on feeder layers of mouse embryonic fibroblasts (*Ma et al., 2022*). Unfortunately, the use of feeder layers precluded the use of this method in a clinical setting and it did not address the mechanistic requirements for the expansion.

PP cells express the components of a large number of signaling pathways, and we hypothesized that a longitudinal transcriptome analysis of non-expanding and occasionally expanding PP cells would provide candidate signaling pathways reasoning that upregulated pathways in expanding cells would promote expansion, whereas downregulated pathways would be blocking expansion and/or favoring differentiation. We leveraged these findings and employed a hypothesis-driven iterative process to define conditions that allowed robust, unlimited expansion, of hPS cell-derived PP cells. We found that the combined stimulation of specific mitogenic pathways, suppression of retinoic acid signaling, and inhibition of selected branches of the TGFβ and Wnt signaling pathways enabled the 2000-fold expansion of PP cells over 10 passages and 40–45 d. Expansion conditions are GMP-compliant and enable the robust, reproducible expansion, as well as intermediate cryopreservation, of PP cells derived from diverse hPS cell lines with essentially identical growth kinetics. These conditions select PDX1$^+$/SOX9$^+$/NKX6-1$^+$ PP cells, suggesting that they will be advantageous for hPS cell lines that may differentiate less efficiently into PP cells. Expanded PP cells differentiated, with very similar efficiency to non-expanded cells, into SC-islet clusters that contained functional β-cells as shown by glucose-stimulated insulin secretion (GSIS) assays.

Our findings will allow the establishment of large banks of PP cells derived under GMP conditions from diverse hPS cell lines, streamlining the generation of SC-islet clusters for further development of the differentiation procedure, diabetes research, personalized medicine, and cell therapies.

## Results

### Identification of candidate signaling pathways implicated in PP expansion

H1 hPS cells were differentiated into PP cells and PP cell expansion was attempted using the initial conditions (CINI) (*Supplementary file 1*) that relied on EGF and FGF2 and A83-01, a broad TGFβ inhibitor of ALK4/5/7 (*Tojo et al., 2005*). The CINI formulation was similar, but not identical, to that used previously in the attempted expansion of PP cells derived through the reprogramming of human fibroblasts into endoderm progenitor cells, where SB431542, another br o ad TGFβ inhibitor of similar specificity (*Inman et al., 2002*), was employed (*Zhu et al., 2016b*). PP cells of comparable quality as assessed by qPCR for the expression of the PP markers *PDX1*, *NKX6-1*, and *SOX9*, as well as for the expression of the liver and gut markers, *AFP* and *CDX2*, respectively, were plated at high density on Matrigel-coated plates and passaged every 4–6 d. In most instances, cell numbers remained either constant or decreased, resulting in growth arrest. Occasionally (one out of five attempts), PP-containing cells (referred to hereafter as just PP cells for simplicity) expanded and could then be maintained for at least up to 10 passages (*Figure 1A*). This was not correlated with higher initial expression of *PDX1*, *NKX6-1*, *SOX9*, or lower expression of *AFP* and *CDX2* (*Figure 1B*). Immunofluorescence experiments and qPCR analyses suggested the maintenance of the PP identity in these expanding PP (ePP) cells since expression of *PDX1*, *NKX6-1,* and *SOX9* remained stable at both the protein (compare *Figure 1—figure supplement 1A–C*) and transcript levels (*Figure 1—figure supplement 1D*). Immunofluorescence analysis of cryosections showed that these ePP cells could subsequently differentiate into endocrine cells (*Figure 1—figure supplement 1E*). These results suggested that EGF, FGF2, and broad TGFβ inhibition were not sufficient for the reproducible expansion of hPS cell-derived PP cells.

PP cells express the components of numerous signaling pathways (*Supplementary file 1*), and we hypothesized that transcriptome comparison of non-expanding and CINI ePP cells could identify pathways implicated in the expansion. Upregulated pathways in ePP cells would be promoting expansion, whereas downregulated pathways would be blocking expansion and/or favoring differentiation. Thus, we compared the RNA-Seq profiles of cells directly after their differentiation into pancreatic progenitors (dPP) (p0) (n = 4), and CINI ePP cells at passage 5 (p5) (n = 3) and p10 (n = 3). Of the analyzed dPP samples, one was subsequently successfully expanded. Analysis for differentially expressed genes (DEGs) identified groups of genes that were continuously up- or downregulated genes and genes showing changes only within the first five passages (*Figure 1—figure supplement 1F*, *Supplementary file 1b*). Metric multidimensional scaling (MDS) plot of the Euclidean distances of the samples demonstrated clustering of the samples strictly according to their passage number. The single p0 sample, which subsequently expanded successfully, clustered with the other p0 samples, suggesting that it was not fundamentally different from the samples that failed to expand. These findings suggested a reproducible adaptation of ePP cells to the culture conditions primarily during the first five passages (*Figure 1C*). Gene Set Enrichment Analysis (GSEA) suggested that components of the TGFβ signaling pathway were negatively correlated with expansion, whereas E2F target genes and DNA replication were positively correlated (*Figure 1D*). E2F TFs are indirectly activated by growth signals to regulate multiple cell cycle genes and promote cell proliferation (*Ertosun et al., 2016*; *Rubin et al., 2020*). This raised the possibility that expanding cells upregulate their own growth factors engaging themselves in an autocrine growth loop. On the other hand, all key components of the Hippo pathway were expressed but were not transcriptionally regulated, except *SHANK2*, a negative regulator, and *WWTR1(TAZ)*, a transcriptional regulator (*Meng et al., 2016*; *Supplementary file 1c*). To gain a better mechanistic insight into potentially involved signaling pathways, we then examined the expression kinetics of all individual ligands and receptors expressed in dPP and ePP cells.

High expression of TGFβ ligands as well as type I and II receptors suggested that all three branches of this signaling pathway were active at p0 (*Supplementary file 1c*). The negative correlation of the TGFβ signaling pathway with the expansion appeared to retrospectively justify the use of the broad TGFβ signal inhibitor A83-01 in the CINI expansion medium (*Tojo et al., 2005*). However, since this

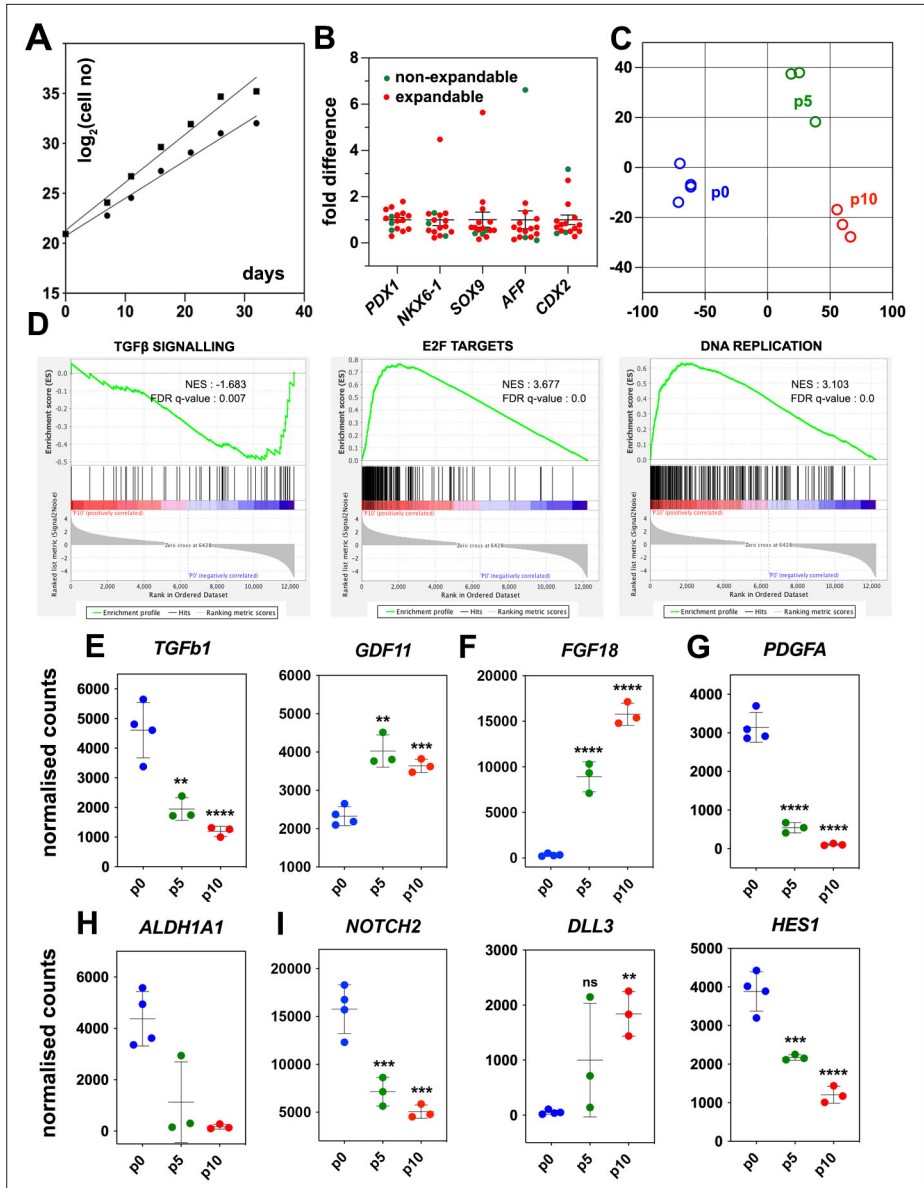

**Figure 1.** Regulated genes and signaling pathways during the expansion of pancreatic progenitor (PP)-containing cells (PP cells) under the initial condition (CINI). (**A**) Growth curves of two samples showing exponential expansion of PP cells for 32 d. (**B**) Expression of pancreatic (*PDX1*, *NKX6-1*, *SOX9*) as well as liver and gut markers (*AFP*, *CDX2*, respectively) at the PP stage of representative differentiations before expansion. Expression at each sample is shown as a fold difference from the average expression level. In red are shown samples that could not be expanded and in green samples that could be expanded. (**C**) Metric multidimensional scaling (MDS) plot representing the Euclidean distance of samples at p0 (n = 4), p5 (n = 3), and p10 (n = 3). (**D**) Enrichment plots for regulated genes (p0 vs p10) of the TGFβ signaling pathway, E2F target genes, and DNA replication show a negative correlation of the TGFβ pathway but a positive correlation of E2F target genes and DNA replication with expansion. (**E–I**) Transcript levels expressed in normalized RNA-Seq counts of p0 (n = 4), p5 (n = 3), and p10 (n = 3) PP cells for genes encoding signals or receptors of the TGFβ (**E**), FGF (**F**) and PDGF (**G**) signaling pathways, the RA-producing enzyme ALDH1A1 (**H**), as well as components of the NOTCH (**I**) signaling pathway. Dots in all graphs represent values from independent experiments. Horizontal lines represent the mean ± SD. Statistical tests were one-way ANOVA using p0 as the control condition for the comparison with *p≤0.033, **p≤0.002, ***p≤0.0002, and ****p≤0.0001.

The online version of this article includes the following figure supplement(s) for figure 1:

**Figure supplement 1.** Regulated genes and signaling pathways during pancreatic progenitor (PP) expansion under initial condition (CINI).

pathway has several branches often with cell-dependent opposing functions, we assessed the expression kinetics of all expressed TGFβ ligands and receptors. The most highly expressed TGFβ ligands, in dPP cells, were *TGFb1* and *BMP2*, the expression of which was repressed nearly fourfold during expansion (*Figure 1E*, *Figure 1—figure supplement 1G*). TGFb1 and BMP2 act through the ALK1/5 and ALK3/6/2 receptors, respectively (*Brown and Schneyer, 2010*), so A83-01 would block TGFB1 but not BMP2 signaling. On the other hand, *ALK4* and its ligand, *GDF11*, remained strongly expressed throughout expansion; *GDF11* was even upregulated (*Figure 1E*, *Supplementary file 1c*), suggesting a possible positive role of this TGFβ signaling branch in the expansion of pancreas progenitors. These findings suggested that the use of broad TGFβ inhibitors such as A83-01 or SB431542 may not be optimal because they do not inhibit BMP2 signaling, whereas they block ALK4 signaling, which might promote PP expansion through GDF11. This provided a mechanistic rationale for the subsequent use of more specific TGFβ inhibitors.

To account for the positive correlation of the expression of E2F targets with expansion, we then examined the gene expression kinetics of growth factors during expansion. There was a striking, nearly 30-fold, upregulation of *FGF18* expression (*Figure 1F*, *Supplementary file 1c*), suggesting a strong requirement for the activation of the MAPK pathway. Incidentally, FGF18 has a selective affinity for FGFR3 and 4 (*Zhang et al., 2006*), the two most highly expressed *FGFRs* in dPP and ePP cells. Other *FGFs* were weakly expressed and not upregulated during expansion (*Supplementary file 1c*). Expression of *FGF18*, as well as of other *FGFs* and *FGFRs*, was detected at several stages of pancreas development in whole pancreata (*Dichmann et al., 2003*). Another potentially interesting growth factor signaling pathway was the PDGF signaling pathway. PDGFR signaling has not been so far implicated in pancreas development or PP expansion but it promotes the expansion of young β-cells (*Chen et al., 2011*). *PDGF* receptors *A* and *B* were stably and strongly expressed during expansion, but the initially strong expression of two of the expressed ligands, PDGFA and B, was downregulated by 30- and 15-fold, respectively (*Figure 1G*, *Figure 1—figure supplement 1H*, *Supplementary file 1c*). These findings provided a mechanistic rationale to provide FGF18 during expansion and/or inhibit PDGF signaling in subsequent experiments.

RA promotes the differentiation of PPs during development (*Lorberbaum et al., 2020*; *Martín et al., 2005*; *Oström et al., 2008*; *Vinckier et al., 2020*). PP cells highly expressed the retinol dehydrogenase 11 (*RDH11*) and aldehyde dehydrogenase 1a1 (*ALDH1A1*) genes, which encode enzymes that convert vitamin A into retinoic acid, as well as the RA nuclear receptors *RARA*, *RARG*, *RXRA*, and *RXRB*. Therefore, in the presence of vitamin A, this pathway could act in an autocrine manner to promote differentiation. Strikingly, successful expansion was accompanied by a dramatic, 25-fold, downregulation in the expression of the RA-producing enzyme *ALDH1A1* (*Figure 1H*, *Supplementary file 1c*). Thus, we speculated that PP cells are poised to initiate RA-mediated differentiation in an autocrine feed-forward loop and, therefore, eliminating vitamin A in the medium might stabilize the PP state.

The Notch signaling pathway is implicated in multiple aspects of pancreatic development including the expansion of PPs and lineage selection (*Afelik et al., 2012*; *Apelqvist et al., 1999*; *Murtaugh et al., 2003*; *Qu et al., 2013*; *Seymour et al., 2020*; *Shih et al., 2012*). The expression of Notch receptors, ligands, and effectors, such as HES1, displayed a complex pattern of changes during expansion. (*Figure 1I*, *Figure 1—figure supplement 1I*, *Supplementary file 1c*) that suggested an overall attenuation, but not silencing, of the pathway.

In summary, the findings suggested that several mechanisms may contribute to stabilizing the PP state. Alone or in combination, inhibition of selected branches of the TGFβ pathway, attenuation of the Notch pathway, additional mitogenic stimulation with FGF18, and suppression of RA and PDGF signaling may lead to reproducible expansion of the PP cells.

## Elimination of RA and selective TGFβ inhibition allow reproducible expansion of PP cells

We assessed several expansion culture conditions (summarized in *Supplementary file 1a*), based on the mechanistic insights discussed above, using PP cells generated with an adaptation of published procedures (*Balboa et al., 2022*; *Mahaddalkar et al., 2020*; *Rezania et al., 2014*; *Shi et al., 2017*; *Supplementary file 1d*). Our strategy to achieve reproducible and robust PP expansion involved activation or inhibition of specific signaling pathways, either individually or in combination. We first

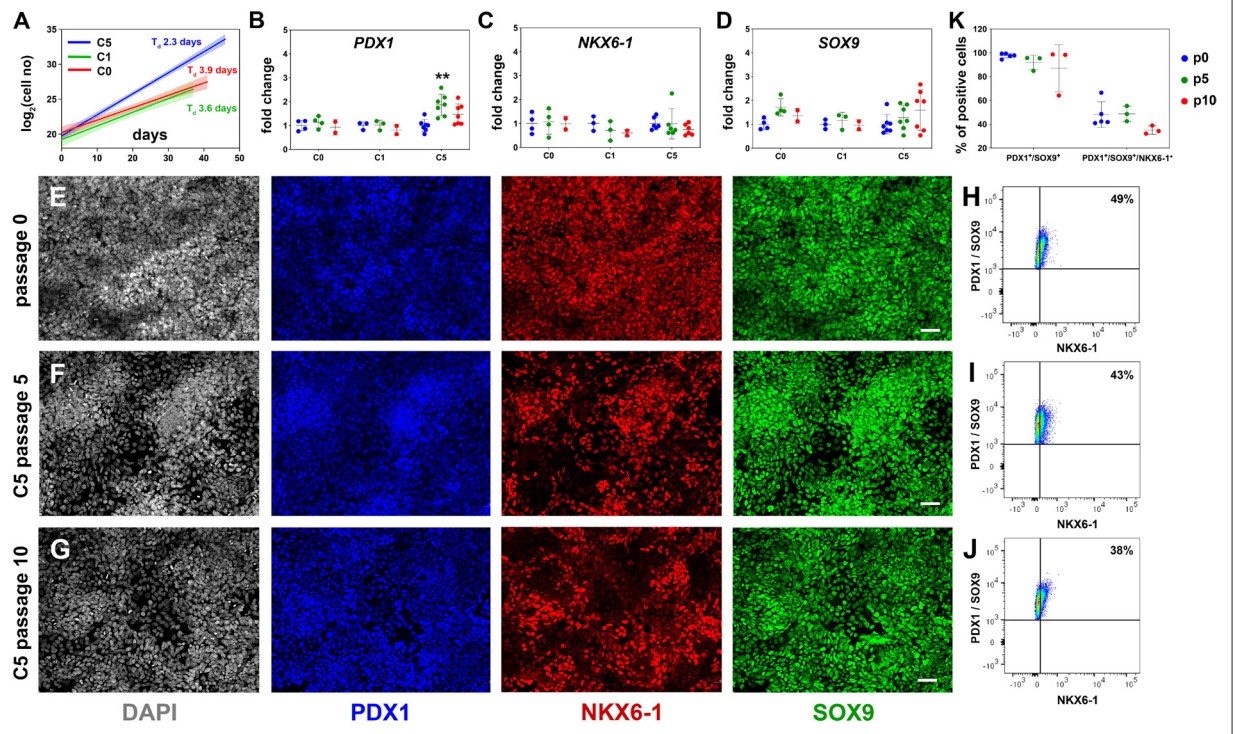

**Figure 2.** Reproducible expansion of pancreatic progenitor (PP) cells under condition 5 (C5). (**A**) Growth curves and regression analysis for PP cells expanded under C0, C1, and C5 for at least 10 passages. The doubling time ($T_d$) of C5-expanded cells (n = 7) was 2.3 d with a 95% confidence interval (CI) of 2.13–2.51 d. This was clearly increased compared to C0- (n = 2, $T_d$ = 3.92 d, 95% CI = 3.22–4.98 d) and C1-expanded cells (n = 2, $T_d$ = 3.55 d, 95% CI = 2.88–4.62 d). The translucent shading represents the 95% CI of the growth rate at the different conditions. (**B–D**) Gene expression profile of C0-, C1-, and C5-expanded cells as shown by qPCR for expression of the key pancreas progenitor markers *PDX1* (**B**), *NKX6.1* (**C**), and *SOX9* (**D**) during the expansion. Expression is normalized against the expression of each marker at p0. (**E–G**) Representative images of immunofluorescent staining of p0 PP cells (**E**) as well as C5-expanded cells at p5 (**F**) and p10 (**G**) for the PP transcription factors PDX1, NKX6.1, and SOX9. (**H–J**) Flow cytometry analysis of p0 PP cells (**H**), as well as C5-expanded cells at p5 (**I**) and P10 (**J**) for PDX1, NKX6.1, and SOX9. (**K**) Cumulative results of the flow cytometry analyses for PDX1+/SOX9+ and PDX1+/SOX9+/NKX6.1+ C5-expanded PP cells at p0, p5, and p10. Dots in all graphs represent values from independent experiments. Horizontal lines represent the mean ± SD. Statistical tests were two-way ANOVA with Tukey's test using p0 as the control condition for the comparison with *p≤0.033, **p≤0.002, ***p≤0.0002, and ****p≤0.0001. Scale bar corresponds to 50 µm.

The online version of this article includes the following figure supplement(s) for figure 2:

**Figure supplement 1.** Reproducible expansion of pancreatic progenitor (PP) cells under condition 5 (C5).

suppressed RA signaling by substituting the vitamin A-containing B27 supplement with a vitamin A-free B27 formulation. Additionally, we substituted A83-01 with the ALK5 II inhibitor (ALK5i II) that targets primarily ALK5, and to a lesser extent ALK3/6, but not ALK4 (*Gellibert et al., 2004*). This was named condition 0 (C0). C0 was further elaborated by replacing FGF2 with FGF18 (C1) as well as adding the Notch inhibitor XXI (*Seiffert et al., 2000*) (C2), or the PDGFR inhibitor CP673451 (*Roberts et al., 2005*) (C3), or adding both Notch and PDGFR inhibitors (C4). Since FGF2 and FGF18 belong to different FGF subfamilies and have overlapping, but not identical, FGFR specificities (*Zhang et al., 2006*), we also addressed a possible synergistic effect of these FGFs in PP expansion by combining them in C5. In contrast to CINI, three of these conditions, C0, C1, and C5, resulted in the reproducible expansion of PP cells for at least 10 (C0, C1, n = 3) or 11 (C5, n = 5) passages with doubling times ($T_d$) of 3.9, 3.6, and 2.3 d, respectively (*Figure 2A*, *Supplementary file 1a*).

Assessment of the expression of key PP genes by qPCR at p5 and p10 suggested that, in all three conditions, initial high levels of *PDX1*, *NKX6-1*, and *SOX9* expression were maintained (*NKX6-1*, *SOX9*) or even transiently increased at p5 (C5, *PDX1*) (*Figure 2B–D*). Expression of *PTF1A* was dramatically decreased in all three conditions, suggesting a shift to a bipotent endocrine/duct progenitor identity (*Figure 2—figure supplement 1A*). Expression of *FOXA2*, involved in PP cell maintenance and endocrine lineage development (*Gao et al., 2010*; *Gao et al., 2008*; *Lee et al., 2005*; *Lee et al., 2019*), was upregulated in C5, indicating that cells expanded in this condition might be more amenable to

terminal endocrine differentiation (*Figure 2—figure supplement 1B*). On the other hand, there was a statistically significant increase in the expression of the liver and gut markers *AFP* and *CDX2* in C0 and C1 and a similar, but weaker, trend in C5 (*Figure 2—figure supplement 1C and D*). This suggested that these conditions (C0, C1, C5) allowed cells to express aspects of liver and gut programs. Since C5 had a significantly lower $T_d$ and a less pronounced increase in *AFP* and *CDX2* expression, we concentrated on the analysis and further improvement of C5.

To first confirm the C5 qPCR results at the protein level, we assessed the expression of several markers by immunofluorescence for C5 ePP cells. Immunofluorescence suggested that PDX1 and SOX9 were uniformly expressed at p0, p5, as well as p10 and that a large number of PP cells were NKX6-1$^+$ at all three different time points even though expansion appeared to somewhat reduce the number of NKX6-1$^+$ cells. (*Figure 2E–G*). Similarly, FOXA2 remained widely expressed at p0, p5, and p10 (*Figure 2—figure supplement 1E–G*). Expression of both AFP and CDX2 appeared to increase transiently upon expansion, at p5 (*Figure 2—figure supplement 1H–J*). We then quantified the expression of the key PP markers PDX1, SOX9, and NKX6-1 by flow cytometry at p0, p5, and p10. In these analyses, PDX1$^+$/SOX9$^+$ cells were first identified based on the gates set by the corresponding control stainings and, then, the number of NKX6-1$^+$ cells was determined in these pre-gated cells based on the gate set by the corresponding control staining (*Figure 2—figure supplement 1K–N*). These experiments confirmed the immunofluorescence experiments showing that at least 90% of all cells were PDX1$^+$/SOX9$^+$ at all passages examined (*Figure 2—figure supplement 1K–N*), whereas nearly 50% of the cells at p0 and p5 and nearly 40% at p10 were PDX1$^+$/SOX9$^+$/NKX6-1$^+$. There was a small, apparent progressive drop in the number of PDX1$^+$/SOX9$^+$ as well as the number of PDX1$^+$/SOX9$^+$/NKX6-1$^+$ cells that did not reach statistical significance (*Figure 2H–K*, *Figure 2—figure supplement 1K–N*).

These experiments established that C5 allowed reproducible and robust PP expansion, the basis of which was the elimination of RA signaling and selective TGFβ inhibition. Additionally, EGF, FGF2, and FGF18 synergized to promote a more robust expansion.

## Expansion conditions promote primarily the proliferation of PP cells

To understand the mechanism behind the successful and reproducible expansion, we asked whether it was due to enhanced cell survival, proliferation, or a combination of both. De novo-generated PP cells were expanded in either C5 or CINI and assayed at p3 for EdU incorporation and apoptosis. The EdU incorporation analysis revealed that C5 cultures contained significantly more EdU$^+$ expanding cells (10.4 ± 1.2%) than CINI (4.2 ± 0.6%) cultures (n = 4) (*Figure 3A–C*, *Figure 3—figure supplement 1A*). The apoptosis assays showed that even though the percentage of 7-AAD$^+$/Annexin V$^+$ cells appeared higher in CINI-expanded cells (22.1 ± 5.0%) than in C5-expanded cells (17.2 ± 4.7%) this difference did not reach statistical significance (n = 6) (*Figure 3C*, *Figure 3—figure supplement 1B and C*).

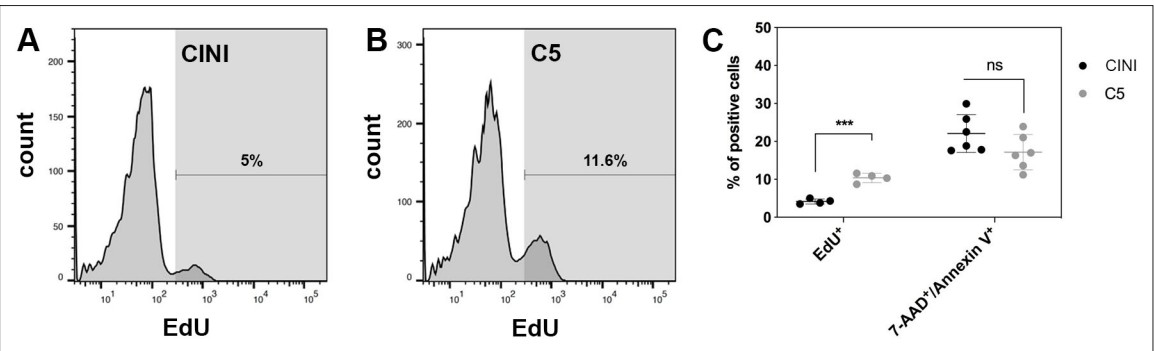

**Figure 3.** Expansion of pancreatic progenitor (PP) cells promotes primarily their proliferation rather than their survival. (**A, B**) Histogram plots showing the percentage of PP cells that had incorporated EdU during expansion under CINI (**A**) and C5 (**B**). (**C**) Summary of flow cytometry data comparing proliferation, measured by EdU incorporation (n = 4), and cell death, measured by Annexin V/7-AAD staining (n = 6), of PP cells expanded under CINI and C5. Dots in all graphs represent values from independent experiments. Horizontal lines represent mean ± SD. Means were compared with multiple *t*-tests and significance is *p≤0.033, **p≤0.0021, ***p≤0.0002, or ****p≤0.0001.

The online version of this article includes the following figure supplement(s) for figure 3:

**Figure supplement 1.** Expansion of pancreatic progenitor (PP) cells promotes primarily their proliferation rather than their survival.

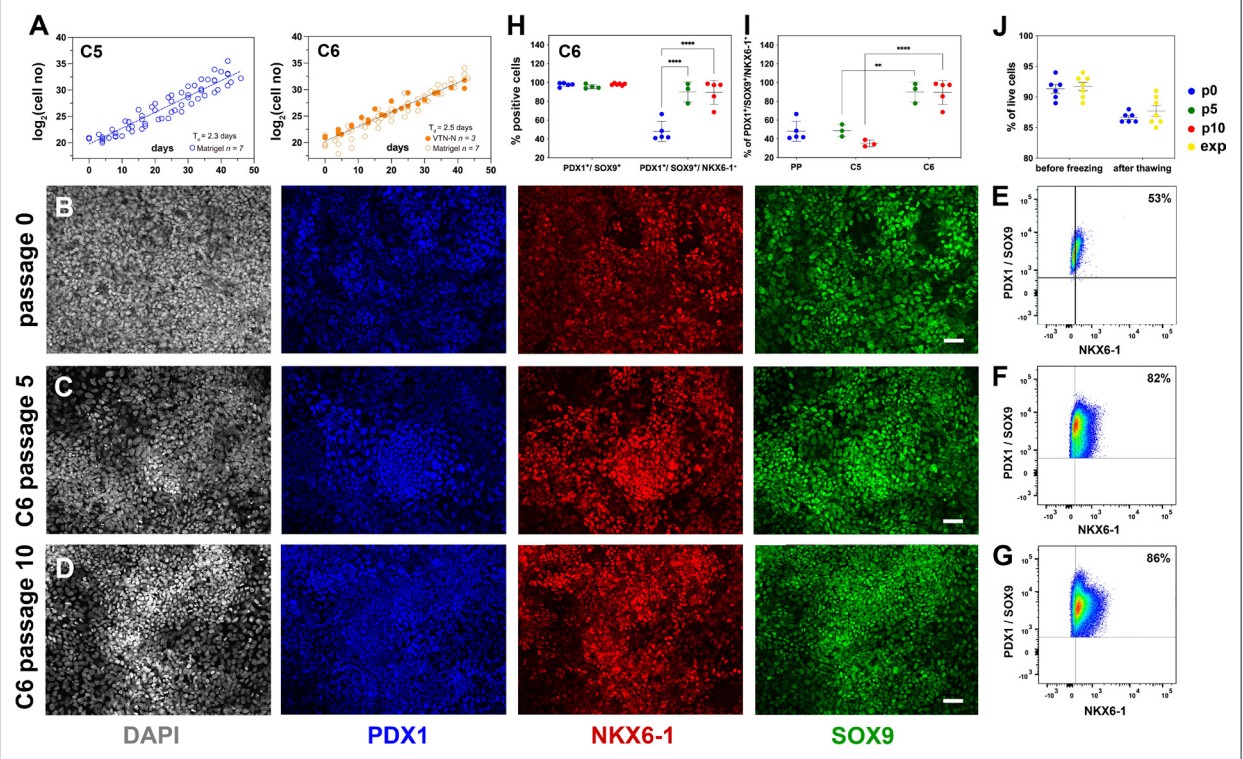

**Figure 4.** Reproducible expansion under condition 6 promotes pancreatic progenitor (PP) identity. (**A**) Growth curves and regression analysis for PP cells expanded under C5 and C6 for 10 passages. The regression line for C5 showed a doubling time of 2.3 d (n = 7) compared to 2.5 d (n = 10) for C6. (**B–D**) Representative images of immunofluorescent staining of p0 PP cells (**B**) as well as C6-expanded cells at p5 (**C**) and p10 (**D**) for the PP transcription factors PDX1, NKX6.1, and SOX9. (**E–G**) Flow cytometry analysis of non-expanded p0 PP cells (**E**) and C6-expanded cells at p5 (**F**) and p10 for PDX1$^+$/SOX9$^+$/NKX6.1$^+$ cells (**G**). (**H, I**) Cumulative results of the flow cytometry analyses for PDX1$^+$/SOX9$^+$ and PDX1$^+$/SOX9$^+$/NKX6.1$^+$ C6-expanded cells at p0, p5, and p10 (**H**) and comparison of the percentage of C5- and C6- expanded PDX1$^+$/SOX9$^+$/NKX6.1$^+$ cells at p5 and p10 (**I**). (**J**) Survival rates of PP cells before freezing and after thawing at p0 or during expansion. Dots in all graphs represent values from independent experiments. Horizontal lines represent the mean ± SD. Statistical tests were two-way ANOVA with Tukey's test using p0 as the control condition for the comparison with *p≤0.033, **p≤0.002, ***p≤0.0002, and ****p≤0.0001. Scale bar corresponds to 50 μm.

The online version of this article includes the following figure supplement(s) for figure 4:

**Figure supplement 1.** Reproducible expansion in condition 6 promotes pancreatic progenitor (PP) identity.

Therefore, C5, compared to CINI, promotes primarily proliferation rather than survival of PP cells, but it is important to note that cell death also appears higher in CINI cells. It is likely that the combination of these effects results in reproducible expansion under C5.

## Canonical Wnt inhibition restricts the upregulation of hepatic fate and promotes PP identity

The upregulation of *AFP* and *CDX2* in C5 ePP cells suggested a drift toward hepatic and intestinal fates that would hinder the efficiency of differentiation into endocrine cells and subsequent maturation (*Nair et al., 2019*). During pancreas development, non-canonical Wnt signaling specifies bipotent liver/pancreas progenitors to pancreatic fates, whereas canonical Wnt signaling leads to liver specification and the emergence of gastrointestinal identity (*Schaffer et al., 2013*; *Muñoz-Bravo et al., 2016*; *Rodríguez-Seguel et al., 2013*). PP cells, at p0 as well as subsequent passages, strongly and stably expressed several *WNT* receptors, co-receptors, as well as canonical and non-canonical signals (*Supplementary file 1c*). Thus, to suppress the upregulation of hepatic and/or intestinal fates, we supplemented C5 with the canonical Wnt inhibitor IWR-1 to selectively inactivate the canonical Wnt signaling (*Chen et al., 2009*) (condition 6; C6, *Supplementary file 1a*).

C6 ePP cells retained a growth rate similar to C5, showing that IWR-1 supplementation did not significantly affect their growth; their T$_d$ was 2.5 d as opposed to 2.3 d for C5 ePP cells (*Figure 4A*). C6 expansions using vitronectin-N (VTN-N) showed no change in expansion efficiency or promotion of

the PP identity (n = *3*). VTN-N is a defined peptide that can be produced under GMP conditions. Since all other media components can also be produced under GMP conditions, this finding established that this approach is also GMP-compliant (*Figure 4A*).

Relative to C5 ePP cells of the same passage number, C6 ePP cells retained strong expression of *PDX1*, *NKX6-1*, and *SOX9* and similar expression levels of *PTF1A* and *FOXA2* (*Figure 4—figure supplement 1A*). Importantly, the expression of the liver markers examined, *AFP, HHEX,* and *TTR*, was significantly lower, by p10, in the C6 ePP cells compared to C5 ePP cells (*Figure 4—figure supplement 1B*). However, expression of *CDX2* was only marginally reduced (*Figure 4—figure supplement 1A*). The expression of *PDX1, NKX6-1, SOX9,* and *FOXA2* was also examined at the protein level by immunofluorescence, which suggested that their expression was stable and persisted at high levels (*Figure 4B–D*, *Figure 4—figure supplement 1C–E*). Similarly to C5 ePP cells, expression of AFP and CDX2 in C6 ePP cells was detectable by immunofluorescence at p5 but significantly reduced at p10 (*Figure 4—figure supplement 1H*). Flow cytometry quantification of the cells expressing key PP markers at p0, p5, and p10 (*Figure 4E–I*, *Figure 4—figure supplement 1I–L*) showed a significant increase in the number of PDX1$^+$/SOX9$^+$/NKX6-1$^+$ C6 ePP cells from 48% ± 11% (n = 5) at p0 to 90% ± 10% (n = 3) at p5 and 95% ± 5% (n = 5) at p10 (*Figure 4H*). This, along with the reduction in the expression of liver markers, was an additional significant improvement over C5 ePP cells at both p5 and p10 (*Figure 4I*). C6 ePP cells could be cryopreserved and recovered with high survival rates (>85%) with no apparent loss of proliferative capacity (*Figure 4J*). Chromosomal stability was also assessed after 16 passages analyzing the G-banding of at least 20 metaphases, and no alterations were found (*Figure 4—figure supplement 1M*).

To further repress the expression of liver and gut markers, we reconsidered BMP inhibition. ALK5i II is considered a relatively weak inhibitor of ALK3 (*Gellibert et al., 2004*). However, BMP2, a ligand of ALK3, was significantly downregulated in the CINI ePP cells (*Figure 1—figure supplement 1G*, *Supplementary file 1c*). Attempting to effectively inhibit ALK3, we substituted ALK5i II with LDN-193189, which inhibits ALK3 with higher potency (*Sanvitale et al., 2013*). LDN-193189 was used alone (C7) or in combination with IWR-1 (C8); however, none of these conditions was efficient in PP cell expansion (*Supplementary file 1a*).

In summary, C6, featuring inhibition of the canonical Wnt signaling by IWR-1, promoted the selection of PDX1$^+$/SOX9$^+$/NKX6-1$^+$ in the expanding cell population and mitigated the upregulation of liver markers at both the gene and protein expression levels. C6 is a robust, highly reproducible, GMP-compliant procedure suitable for application in cell therapies.

## Expansion under C6 stabilizes PP cell identity by repressing differentiation and alternative cell fates

To understand how the C6 expansion procedure affects the transcriptome of PP cells, we performed RNA-Seq analyses on dPP cells (p0) derived from independent differentiations and the corresponding p5 and p10 ePP cells. Principal component analyses (PCA) showed that the main component, PC1, represented most (81%) of the variance among samples and clearly separated p0 PP cells from either p5 and p10 ePP cells. All expanded cells were clustered remarkably close together on the PC1 axis (*Figure 5—figure supplement 1A*), suggesting rapid stabilization of the ePP transcriptome after just five passages. This was confirmed by correlation analyses of the transcriptome profiles showing that all major changes occurred between p0 and p5 (*Figure 5A and B*, *Figure 5—figure supplement 1B*). Results from Gene Ontology (GO) and Kyoto Encyclopedia of Genes and Genomes (KEGG) analyses of DEGs between p0 and p10 were consistent with a cell adaptation to culture conditions, the signaling molecules used for the expansion, and an effect on the differentiation process (*Figure 5C*, *Figure 5—figure supplement 1C*). We then compared the transcriptome of our VTN-N ePP cells with that of ePP cells expanded on feeders (*Ma et al., 2022*), on fibronectin (FN) (*Nakamura et al., 2022*), as well as with that of human fetal PPs (*Ramond et al., 2018*). Initial PCA suggested that all in vitro-derived PP cells clustered away from fetal cells (*Figure 5—figure supplement 1D*). Thus, we subsequently restricted subsequent comparisons only among in vitro-derived PP cells.

In comparative PCA, with the other in vitro-derived ePP cells (*Ma et al., 2022*; *Nakamura et al., 2022*), our ePP cells clustered separately with very high similarity among p5 and p10 ePP cells (*Figure 5D*). To identify the molecular basis of this difference, we used a variance stabilizing transformation (vst) function (*Love et al., 2014*) and hierarchical clustering to identify genes that

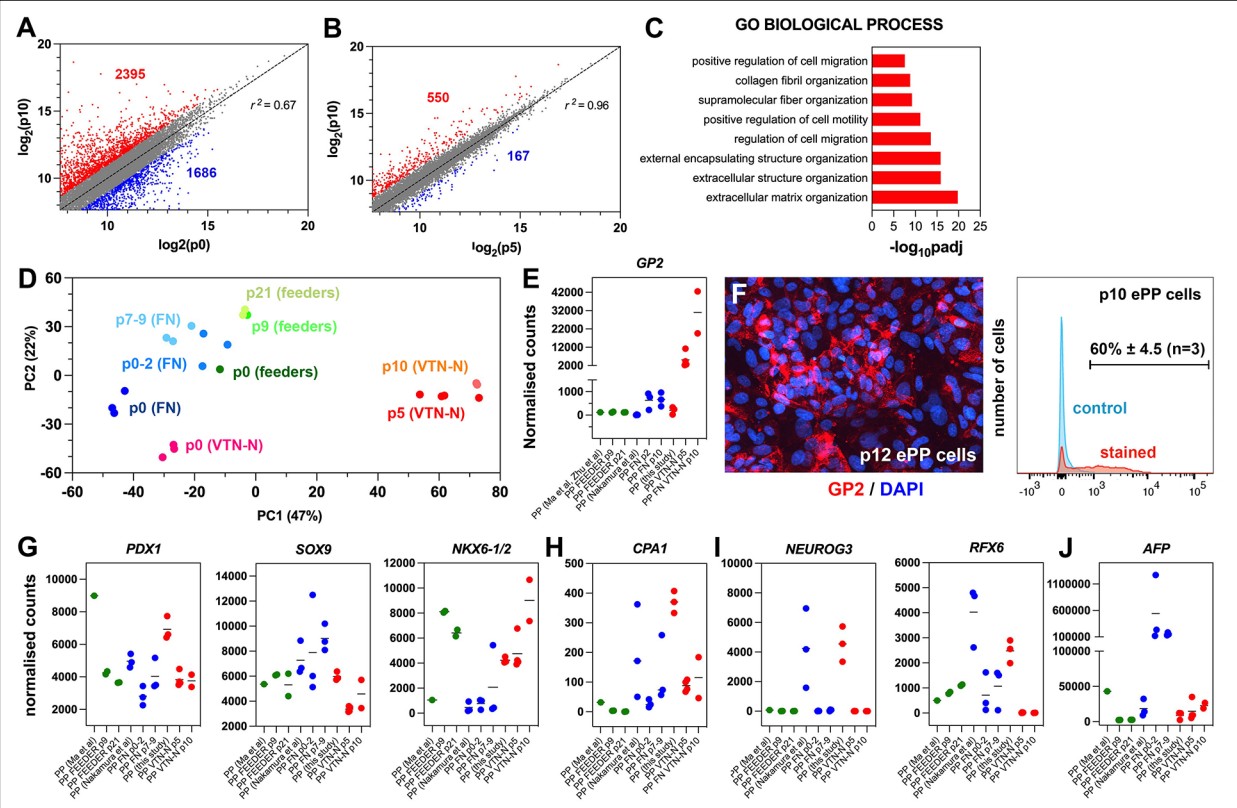

**Figure 5.** Expansion under C6 stabilizes pancreatic progenitor (PP) cell identity by repressing differentiation and alternative cell fates. (**A, B**) Correlation analyses of the transcriptome profiles of non-expanded (p0) and p10 expanded PP cells (**A**) and the transcriptome profiles of p5 and p10 expanded cells (**B**). The numbers of upregulated and downregulated genes (normalized counts ≥200 and 0.5≥ FC ≥ 2) are shown in red and blue, respectively, and r is the correlation coefficient. (**C**) Mostly affected biological processes between p0 and p10. (**D**) Principal component analysis (PCA) of feeder expanded cells and corresponding p0 cells (shades of green), fibronectin (FN)-expanded cells, and corresponding p0 cells (shades of blue), as well as vitronectin-N (VTN-N)-expanded cells and corresponding p0 cells (shades of red). Darker shades correspond to earlier passages. (**E**) Comparative expression levels of *GP2* in normalized RNA-Seq counts. (**F**) GP2 immunofluorescence of p12 expanded PP cells and FC of p10 expanding PP (ePP) cells. (**G–J**) Comparative expression levels of progenitor (**G**), multipotent progenitor cell (MPC) (**H**), endocrine (**I**), and liver (**J**) markers in normalized RNA-Seq counts of ePP cells and their corresponding p0 cells expanded on feeders (green), FN (blue), and VTN-N (red). Dots in all graphs represent values from independent experiments. Horizontal lines in the graphs represent the mean value.

The online version of this article includes the following figure supplement(s) for figure 5:

**Figure supplement 1.** Expansion under C6 stabilizes pancreatic progenitor (PP) cell identity by repressing endocrine differentiation.

are differentially regulated in our ePP cells (*Figure 5—figure supplement 1E*). Interestingly, these consisted only of upregulated genes and GO analyses showed that affected categories referred to epithelial maintenance and differentiation, cellular signaling response and metabolic processes (GO BP), membrane components (GO CC), and metabolism (GO MF and KEGG) (*Figure 5—figure supplement 1F*). An interesting upregulated gene in our ePP cells was *GP2*, a gene encoding a zymogen granule membrane glycoprotein that has been identified as a unique marker of human fetal PPs (*Cogger et al., 2017*; *Ramond et al., 2017*). Importantly, GP2[+]-enriched hPS cell-derived PP cells are more efficiently differentiating into pancreatic endocrine cells (*Aghazadeh et al., 2022*; *Ameri et al., 2017*). Direct comparison of *GP2* expression among ePP cells showed that our expansion method resulted in a progressive strong *GP2* upregulation and expression. This expression was nearly 50-fold higher than expression in the FN ePP cells at p10, whereas feeder ePP cells did not express this marker to any appreciable extent (*Figure 5E*, *Supplementary file 1e*). Robust GP2 expression in our ePP cells was confirmed by immunofluorescence and flow cytometry, where 60% ± 4.5% (n = 3) of the ePP (p10-12) cells scored positive (*Figure 5F*).

To further understand the differences among ePP cells, we compared the expression of several additional pancreatic gene markers. *PDX1* expression was generally reduced after expansion but

remained at comparable levels among the three procedures (*Figure 5G*, *Supplementary file 1e*). *SOX9* expression remained stable in the feeder ePP cells and increased in the FN PP cells, whereas it was downregulated in our ePP cells (*Figure 5G*, *Supplementary file 1e*). Because SOX9 is also a major driver of the ductal pancreatic program, we assessed whether higher SOX9 levels might be associated with higher levels of this program. We examined the collective expression of TFs driving the duct program, such as *PROX1*, *HES1*, *GLIS3,* and *ONECUT1*. During development, these genes are initially expressed in bipotent progenitors and then to duct progenitors and differentiated duct cells but they are excluded from the endocrine compartment (*Bastidas-Ponce et al., 2017*). Consistent with the stronger *SOX9* expression, overall expression levels of these TFs were higher in FN ePP cells suggesting that the ductal program was more active in these cells (*Figure 5—figure supplement 1G*, *Supplementary file 1e*). During pancreas development, both *NKX6-1* and *NKX6-2* are expressed in progenitor cells, acting in concert to define bipotent progenitors and to subsequently specify endocrine cells (*Binot et al., 2010*; *Henseleit et al., 2005*; *Nelson et al., 2007*; *Pedersen et al., 2005*; *Schaffer et al., 2010*). The antibody we used most likely recognizes both proteins, given the very high similarity of NKX6-1 and NKX6-2 at the region of the antigen (*Figure 5—figure supplement 1H*). Consistent with the flow cytometry experiments, our expanded cells showed that the combined *NKX6-1/2* expression was strongly upregulated and similar to that of feeder ePP cells (*Figure 5G*, *Supplementary file 1e*). Interestingly, that was due to a strong, expansion-dependent upregulation of *NKX6-2* (*Figure 5—figure supplement 1I*, *Supplementary file 1e*). Here it is important to note that whereas *NKX6-1/2* expression in the feeder and our ePP cells is strong in all samples analyzed, it is very low in all FN ePP samples with the exception of a single sample (*Figure 5G*, *Supplementary file 1e*). This suggests a lack of reproducibility in the expansion of PDX1$^+$/NKX6-1$^+$ FN ePP cells. The expression of other progenitor TF genes such as *FOXA2* and *RBPJ* remained comparable in all expanded cells, although *FOXA2* retained higher levels in the FN ePP cells and *RBPJ* retained higher levels in our ePP cells (*Figure 5—figure supplement 1I*, *Supplementary file 1e*).

Gene expression analyses, during expansion in the C5 medium, showed complete repression of *PTF1A* expression during the expansion (*Figure 2—figure supplement 1A*). This was also observed in the C6 expansion medium (*Figure 4—figure supplement 1A*). During development, Nkx6-1/Nkx6-2 act antagonistically to Ptf1a to downregulate its expression and promote the conversion of multipotent progenitor cells (MPCs) into bipotent cells (BP) cells (*Schaffer et al., 2010*). Thus, we asked whether PP expansion promoted BP identity (PDX1$^+$/SOX9$^+$/NKX6-1$^+$/PTF1A$^-$ cells) to the expense of MPC (PDX1$^+$/SOX9$^+$/NKX6-1$^+$/PTF1A$^+$) identity. The RNA-Seq analyses showed that *PTF1A* repression was a common feature of all expansion procedures (*Figure 5—figure supplement 1J*, *Supplementary file 1e*). Expression of *CPA1*, a marker of MPCs (*Zhou et al., 2007*), was also reduced during expansion, further supporting a transition to BP identity. This downregulation was notable in the FN and VTN-N-expanded cells because *CPA1* expression of the corresponding p0 cells was substantially higher (*Figure 5H*, *Supplementary file 1e*). We also found that *DCDC2A*, a BP progenitor marker (*Scavuzzo et al., 2018*), was expressed in both FN ePP and VTN-N ePP cells, but not in feeder ePP cells (*Figure 5—figure supplement 1J*, *Supplementary file 1e*). These data suggested that ePP cells, particularly FN and VTN-N ePP cells, resemble BPs rather than MPCs.

NEUROG3$^+$ pancreatic endocrine progenitors divide very rarely, if at all, and employ a feed-forward mechanism for their differentiation (*Azzarelli et al., 2017*; *Krentz et al., 2017*; *Wang et al., 2008*). Therefore, it is expected that an efficient PP expansion procedure would efficiently repress the endocrine program and loss of PP cells by differentiation toward the endocrine lineage. Indeed, a common feature of all expansion procedures was the repression of the endocrine differentiation program. Key TFs such as *NEUROG3* and its downstream effectors *NEUROD1*, *NKX2-2*, and *INSM1* were essentially switched off with very similar efficiency in all PP cells; however, it should be noted that expression of these genes was already very low in p0 cells in the . protocol (*Ma et al., 2022*). On the other hand, expression levels of *RFX3* and *RFX6* were generally higher in the feeder and FN ePP cells (*Figure 5I*, *Figure 5—figure supplement 1K*, *Supplementary file 1e*). Expression of terminal endocrine differentiation markers in all ePP cells was also negligible, particularly at late passages (*Supplementary file 1e*).

Expression of acinar TFs such as *BHLHA15* and *RBPJL* was virtually undetectable in all expanded cells (*Supplementary file 1e*). Finally, we compared the expression of liver and gut markers. Feeder ePP cells retained only negligible expression of the liver markers *AFP* and *HHEX*. However, expression

**Figure 6.** Expansion of H9-derived and CRTD1-derived pancreatic progenitor (PP) cells under C6. (**A, B**) Growth curve and regression analysis of the expansion of H9-derived PP cells (**A**) and CRTD1-derived PP cells (**B**). (**C, D**) Flow cytometry analysis for PDX1$^+$/SOX9$^+$ and PDX1$^+$/SOX9$^+$/NKX6.1$^+$ cells during the expansion under C6 of H9-derived PP cells (**C**) and CRTD1-derived PP cells (**D**) at p0, p5, and p10. Dots in all graphs represent values from independent experiments. Statistical tests were two-way ANOVA with Tukey's test using p0 as the control condition for the comparison with *p≤0.033, **p≤0.002, ***p≤0.0002, and ****p≤0.0001.

The online version of this article includes the following figure supplement(s) for figure 6:

**Figure supplement 1.** Expansion of H9-derived and CRTD1-derived pancreatic progenitor (PP) cells.

of these markers was nearly sevenfold and tenfold higher, respectively, in the FN ePP cells compared to our ePP cells, suggesting an overall decreased propensity to endocrine differentiation (*Figure 5J*, *Figure 5—figure supplement 1L*, *Supplementary file 1e*). Finally, expression of the gut marker *CDX2* was slightly higher in the feeder ePP cells but generally comparable in all three methods (*Figure 5— figure supplement 1L*, *Supplementary file 1e*).

In summary, the comparative transcriptome analyses suggested that ePP cells transition from the MPC to the BP state and that our C6 expansion procedure is more efficient at strengthening the PP identity and efficiently repressing the initiation of endocrine differentiation and alternative liver fate.

## Expansion under C6 selects PDX1$^+$/SOX9$^+$/NKX6-1$^+$ cells derived from either XX or XY hPS cells

Different hPS cell lines may vary in their differentiation efficiency, and this complicates the clinical implementation of this technology. C6 expansion resulted in the selection of PDX1$^+$/SOX9$^+$/NKX6-1$^+$ cells, and this would be advantageous for iPS cell lines that may not differentiate as efficiently. We next assessed whether the C6 expansion condition, established using the male H1 human embryonic stem (hES) cell line, was similarly applicable to other hPS cell lines of either sex. To address this, we used the female H9 hES cell line, a line that has a preference for neural rather than endoderm differentiation, and a male iPS cell line derived in the CRTD (CRTD1, hPSCreg: CRTDi004-A) with unknown lineage preference. Cells were differentiated into PP cells in monolayer culture (*Supplementary file 1d*), and PP cells were then expanded in C6 for at least 10 passages. The comparison of the growth curves of H1- (H1-PP), H9- (H9-PP), and CRTD1- (CRTD1-PP)-derived ePP cells did not suggest statistically significant differences. A $T_d$ of 2.3 and 2.2 d for H9-PP and CRTD1-PP-derived ePP cells, respectively, was calculated. Expansion in VTN-N-coated cell culture surface appeared at least equally efficient as the expansion on Matrigel for H9- and CRTD1- PP cells (*Figure 6A and B*).

We then compared the maintenance of the PP identity during the expansion of H9-PP and CRTD1-PP cells to that of H1-PP cells using qPCR to quantify gene expression levels at p0, p5, and p10. Expression of *PDX1*, *NKX6-1*, and *SOX9* in H9-derived PP cells was strikingly similar to that of H1-derived PP cells at p0 and also following expansion. Expression of these markers in CRTD1-PP cells diverged with decreased *PDX1* expression, at p0 and p10, but increased *NKX6-1* expression at p5 and p10 (*Figure 6—figure supplement 1A–C*). Expression of *FOXA2* and *PTF1A* in p0 H9-PP and CRTD1-PP ePPs was also very similar to that in H1-PP cells with the exception of a transient increase of *FOXA2* expression at p5 (*Figure 6—figure supplement 1D and E*). Expression of *AFP* and *CDX2* was also very similar in H9-PP and CRTD1-PP cells compared to H1-PP cells except for a transient *CDX2* upregulation in CRTD1-PP cells at p5 (*Figure 6—figure supplement 1F and G*).

We then evaluated the presence of PDX1$^+$/SOX9$^+$ and PDX1$^+$/SOX9$^+$/NKX6-1$^+$ cells in H9-PP and CRTD1-PP cells at p0, p5, and p10 using flow cytometry. H9 cells were less efficiently differentiated into PDX1$^+$/SOX9$^+$ cells in comparison to either H1 or CRTD1 cells giving rise to 77% ± 8% (n = 6) as opposed to 98% ± 2% (n = 5) and 89% ± 5% (n = 3) PDX1$^+$/SOX9$^+$ PP cells for H1- and CRTD1-PP cells,

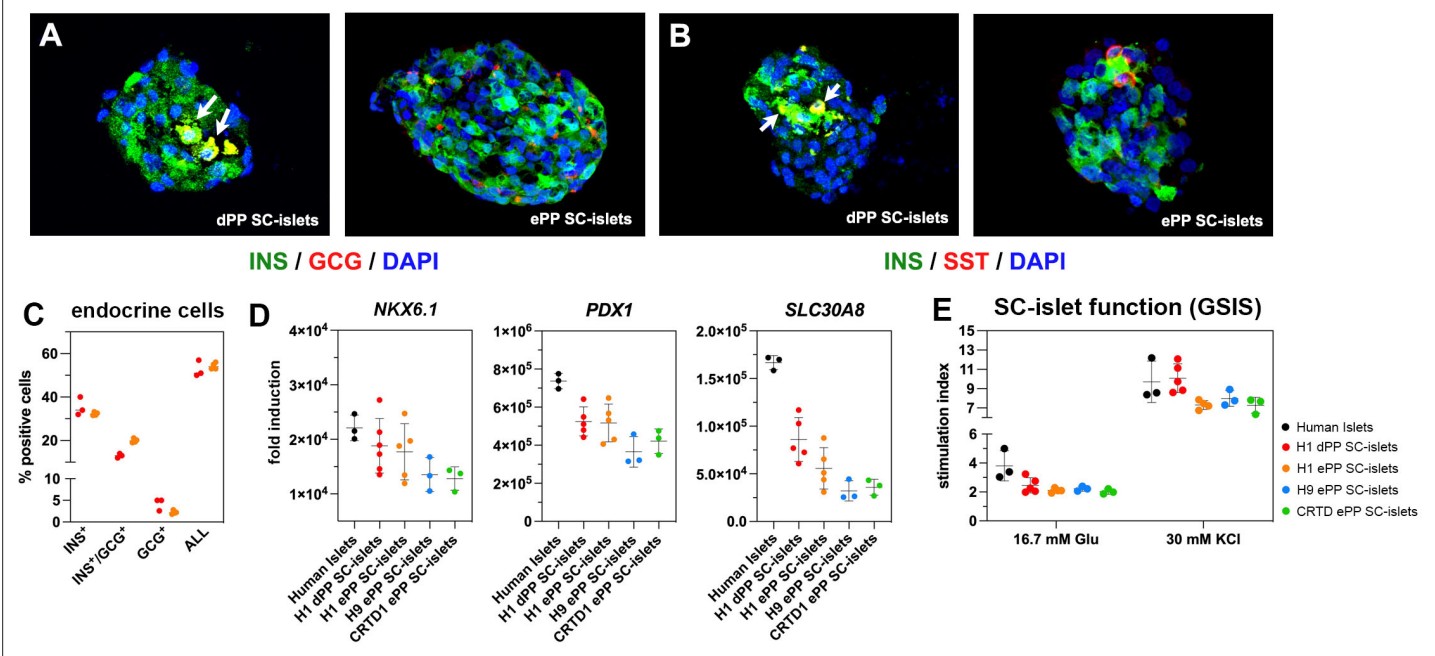

**Figure 7.** Differentiation of expanding pancreatic progenitor (ePP) cells into SC-islets containing functional β-cells. (**A, B**) Immunofluorescence analysis of SC-islets derived from p0 PP cells (dPP) or expanded PP cells for at least 10 passages (ePP) for INS and GCG expression (**A**) or INS and SST expression (**B**). (**C**) Percentages of INS+, INS+/GCG+, as well as GCG+ cells in SC-islets derived from dPP or ePP cells as determined by flow cytometry. (**D**) Expression levels of *NKX6-1*, *PDX1*, and *SLC30A8* as determined by qPCR and expressed as fold induction relative to expression levels in human pluripotent stem (hPS) cells. (**E**) Secretion of C-peptide following sequential stimulation by 16.7 mM glucose and 16.7 mM glucose/30 mM KCl (30 mM KCl) after exposure in basal conditions with 2.8 mM glucose. Stimulation index is the ratio of secretion under these conditions to secretion in basal conditions. Dots in all graphs represent values from independent experiments. Horizontal lines represent the mean ± SD. Statistical tests were two-way ANOVA with Tukey's test using p0 as the control condition for the comparison with *p≤0.033, **p≤0.002, ***p≤0.0002, and ****p≤0.0001.

The online version of this article includes the following figure supplement(s) for figure 7:

**Figure supplement 1.** Differentiation of expanding pancreatic progenitor (ePP) cells into SC-islets containing functional β-cells.

respectively. However, this percentage increased to 91% (n = 3) by p10, similar to that for H1- (98% ± 1%) (n = 5) and CRTD1- (95% ± 6%) (n = 4) p10 ePP cells (*Figures 4H and 6C and D*). Importantly, the percentage of PDX1+/SOX9+/NKX6-1+ H9-PP and CRTD1-PP cells increased from 45% ± 29% and 39% ± 6% at p0 to 90% and 85% ± 6% at p10, respectively. This was a very similar selection to that observed for H1-derived PP cells, which was from 48% ± 11% at p0 to 89% ± 13% at p10 (*Figures 4H and 6C and D*). Chromosomal stability, following expansion, was confirmed also for these lines since no alterations were found after analysis of the G-banding of at least 20 metaphases for each line at p12 (H9-PP cells) or p13 (CRTD1-PP cells) (*Figure 6—figure supplement 1H and I*).

## All C6 ePP cells differentiate into SC-islets containing functional β-cells

Having established that our PP expansion procedure is efficient across PP cells derived from different hPS cell lines, we asked whether ePP cells can be differentiated equally efficiently into SC-islets. H1 dPP cells as well as H1, H9, and CRTD1 ePP cells, expanded for at least 10 passages, were clustered in microwells and differentiated using an adaptation of published media (*Mahaddalkar et al., 2020*; *Rezania et al., 2014*; *Shi et al., 2017*; *Supplementary file 1d*) to generate SC-islets.

Both H1 dPP and ePP cells gave rise to similar clusters containing INS+/NKX6-1+, INS+/MAFA+, GCG+, and SST+ endocrine cells (*Figure 7A and B*, *Figure 7—figure supplement 1A and B*), as well as similar percentages of INS+, INS+/GCG+, and GCG+ cells as determined by flow cytometry (*Figure 7C*, *Figure 7—figure supplement 1C and D*). The total number of INS+ and GCG+ cells was between 50 and 55% in both H1 dPP- and ePP-derived SC-islets (*Figure 7C*). We also compared the differentiation propensity of H1 dPP and ePP cells after their differentiation into SC-islets by evaluating the expression of the duct marker *KRT19*. Its expression was significantly lower in ePP cells, but

it should be noted that in both cases expression was extremely weak (*Figure 7—figure supplement 1E*).

Gene expression levels for differentiated endocrine markers including β-cell markers such as *PDX1*, *NKX6-1*, *SLC30A8* (*Figure 7D*), *INS*, *MAFA,* as well as *GCG* and *SST* (*Figure 7—figure supplement 1F*) were very similar between H1 dPP-derived SC-islets and SC-islets derived from either H1, H9, or CRTD1 ePP cells. Expression of *PDX1*, *NKX6-1*, *SLC30A8,* and *SST* was comparable between human islets and dPP or ePP SC-islets (*Figure 7D*, *Figure 7—figure supplement 1F*), but expression of *INS*, *GCG,* and *MAFA* was substantially higher in human islets reflecting the less advanced maturation of SC-islets (*Figure 7—figure supplement 1F*).

Finally, we assessed the functionality of the β-cells in H1 dPP- as well as H1, H9, and CRTD1 ePP-derived SC-islets in static GSIS assays where SC-islets were sequentially incubated in Kreb's buffer containing low glucose levels (2.8 mM), Kreb's buffer containing high glucose levels (16.7 mM), and finally a depolarizing Kreb's buffer containing high glucose levels (16.7 mM) and KCl (30 mM). As a comparison, human islets were processed in a similar manner. Human C-peptide levels were measured from the supernatant of successive incubations and used to calculate the fold stimulation. These experiments showed that dPP- and ePP-derived SC-islets contained β-cells of very similar functionality (*Figure 7E*) and, as expected, close, but not similar, functionality to human islets. Importantly, the amount of secreted human C-peptide under all different conditions was very similar among dPP and ePP-derived SC-islets (*Figure 7—figure supplement 1G*).

In summary, both dPP and ePP cells, derived from different hPS cell lines, differentiate into SC-islets with essentially the same efficiency and contain β-cells of similar functionality.

## Discussion

The unlimited expansion of progenitor cells holds significant therapeutic promise but remains an important challenge in regenerative medicine. Progenitor cells are dynamic entities, and the key objective in these efforts is to uncouple survival and proliferation from widely employed feed-forward mechanisms that promote their differentiation. The latter are not exclusively regulated by extrinsic signals but rely to a large extent on internal regulators as well as autocrine signals. Thus, the hallmarks of efficient progenitor expansion would be the maintenance of key progenitor features, the efficient suppression of differentiation programs and alternative lineages, and the capacity to efficiently differentiate under appropriate conditions. For therapeutic applications, such expansion should also be efficient, reproducible, applicable across different cell lines and compatible with chemically defined culture media.

Regarding pancreatic development, in particular, several feed-forward networks have been documented with exquisite detail (*Arda et al., 2013*). In the early PPs, a Sox9/Fgf feed-forward loop is essential to escape liver fate and promote pancreas identity and expansion (*Seymour et al., 2012*). Fgf10 was identified as a possible extrinsic signal, but it was shown that it also promotes liver identity (*Hart et al., 2003*; *Jung et al., 1999*; *Norgaard et al., 2003*; *Rossi et al., 2001*). Ptf1a initially forms heterodimers with Rbpj to promote expansion of MPCs and activate transcription of Rbpjl (*Masui et al., 2010*). Expression of PTF1A in the MPCs is essential to set in motion an epigenetic cascade that is required for subsequent duct and endocrine differentiation (*Miguel-Escalada et al., 2022*). Rbpjl eventually replaces Rbpj in its complexes with Ptf1a, and the Ptf1a.Rbpjl complexes promote the specification of acinar progenitors (*Masui et al., 2010*). Their mutual cross-repression with Nkx6-1/2 defines the acinar progenitors and BPs, respectively (*Schaffer et al., 2010*). We found that PP cells at p0 resemble MPCs because they express high levels of *PTF1A* but also *CPA1*, a marker of MPCs (*Zhou et al., 2007*). Expansion promotes the switch to BP identity as documented by the strong downregulation of both *PTF1A* and *CPA1* and the corresponding upregulation of *NKX6-1/2* as well as *DCDC2A* (*Scavuzzo et al., 2018*), suggesting that ePP cells may resemble bipotent progenitors as described in mice (*Schaffer et al., 2010*).

Sphingosine-1-phosphate signaling activates YAP to set the stage for the endocrine differentiation of PP cells (*Serafimidis et al., 2017*), and the YAP/TEAD complex regulates the enhancer network that maintains PP cells in a proliferative state (*Cebola et al., 2015*). In BP cells, YAP activation mediates a mechanotransduction pathway to maintain cells in a proliferative state and its inactivation is a prerequisite for endocrine differentiation (*Cebola et al., 2015*; *Rosado-Olivieri et al., 2019*). The latter mechanism was exploited to improve the in vitro pancreatic β-cell differentiation (*Hogrebe*

*et al., 2020*). As expected, gene expression of all the Hippo pathway components is strong in PP cells, but, even though some of them are differentially regulated, there is not a clearly identifiable pattern to ascribe pathway activation or inactivation, at least at the gene expression level, during expansion. Upon YAP inactivation and endocrine commitment, the TFs Myt1 and Neurog3 form a feed-forward loop to promote the final commitment into the endocrine lineage (*Wang et al., 2008*). Differentiating cells are dynamic cell populations that do not move synchronously through the successive differentiation stages, and this can explain the substantial expression levels of the TFs of the endocrine program in PP cells. The expansion procedure we propose was successful in freezing cells in a state preceding endocrine commitment as shown by the dramatic repression of all endocrine TFs.

We hypothesized that the expression kinetics of signals and signal receptors during expansion under CINI would give us indications on which signaling pathways should be either repressed or activated to promote reproducible expansion. This approach was successful, and we favor the hypothesis that these are autocrine signals and receptors expressed in the PP cells themselves because this modulation remains necessary even at late passages when the percentage of PDX1+/SOX9+/NKX6-1+ is around 90%. However, we cannot exclude the possibility that some or even all of the inhibited signals, including the suppression or RA signaling through the withdrawal of vitamin A, may come from non-PP cells present in the culture. In this case, their action would be to block the expansion and/or survival of alternative cell fates. A convincing answer to this question should await single-cell RNA-Seq analyses, but it is worth noting that a high initial PP cell density was an essential requirement for robust early expansion for all cell lines. This suggests that autocrine signals play a key role in the process.

We found that a combination of EGF, FGF2, and FGF18 promoted robust expansion of hPS cell-derived PP cells. EGF promotes *NKX6-1* activation (*Nostro et al., 2015*), but, on its own, it could not support consistent PP cell expansion (CINI, *Supplementary file 1a*). FGF2 is a widely used mitogen, and whereas it enhanced PP proliferation (C1) it was less efficient than FGF18 (C5, *Supplementary file 1a*). There was a clear synergy of the two mitogens in promoting expansion and possibly the enrichment in NKX6-1+ cells (C6, *Supplementary file 1a*). NKX6-1 regulates multiple cell cycle genes (*Taylor et al., 2015*), and thus, some of the effects of these mitogens might be indirectly reinforced through NKX6-1. The comparison of early (p3) CINI and C5 cells suggested that higher proliferation in the latter was the main driver of successful expansion. Cell death appeared also lower in C5 cells (17.2 ± 4.7% against 22.1 ± 5.0%), and thus, it is likely that the combination of these effects results in reproducible expansion under C5. The high rates of cell death suggest that committed non-PP cells are primarily affected and bona fide PP cells as well as cells still competent to become bona fide PP cells compensate with higher proliferation. This question could be resolved by co-staining the cells with PP markers as well as cell death and proliferation markers.

PP cells express the enzymes necessary to convert vitamin A into retinoic acid (RA) (*Supplementary file 1c*), which, in turn, promotes differentiation of the pancreas progenitors (*Lorberbaum et al., 2020*; *Martín et al., 2005*; *Oström et al., 2008*; *Vinckier et al., 2020*). Thus, eliminating vitamin A from the expansion medium was important to uncouple PP expansion from differentiation. The TGFβ signaling pathway plays a complex role in the induction, maintenance, and endocrine differentiation of pancreas progenitors (*Guo et al., 2013*; *Sanvito et al., 1994*; *Spagnoli and Brivanlou, 2008*; *Tulachan et al., 2003*), and this was reflected in the widely variable gene expression kinetics of receptors and ligands during CINI expansion. Accordingly, our experiments suggested that the highly specific ALK5i II inhibitor (*Sanvitale et al., 2013*) (C5, C6) was more efficient than the less specific A83-01 inhibitor (*Tojo et al., 2005*) (CINI) in promoting PP expansion. Further inhibition of the pathway with the addition of LDN193198, which targets ALK3 more efficiently than ALK5i II (*Gellibert et al., 2004*), dramatically reduced PP expansion (C7, *Supplementary file 1c*). We documented that preferential expansion of progenitors, rather than cell survival, was the main mechanism of the expansion, in agreement with the idea that the expansion medium effectively decoupled proliferation from differentiation. Finally, the addition of the canonical Wnt inhibitor IWR-1 (*Chen et al., 2009*) significantly reduced AFP expression and promoted an efficient selection of NKX6-1+ cells, without affecting the proliferation rate, presumably by reducing the branching of expanding cells toward the hepatic fate.

The molecules employed in our C6 expansion medium are not, with the partial exception of FGF-18, foreign to the differentiation conditions used to generate SC islets. RA has been used to induce

the primitive gut tube conversion to PP cells, and ALK5i II has been used to promote the PP to endocrine progenitor conversion (*Hogrebe et al., 2021*; *Rosado-Olivieri et al., 2019*) Additionally, IWR-1 has been proposed to increase endocrine differentiation (*Sharon et al., 2019*). This may appear contradictory, but we think that it is the combination of signals that promotes a certain cell state rather than specific signals. Accordingly, we carried out the generation of PP cells before expansion and their differentiation after expansion in exactly the same way as for PP cells that were not expanded. In these procedures, we employed RA and ALK5i II as published earlier, but in our hands, IWR-1 did not make a difference in either the dPP or ePP to endocrine progenitor transition.

Culture conditions to expand hPS cell-derived PP cells have recently been reported (*Ma et al., 2022*; *Nakamura et al., 2022*) but suffer from several disadvantages as outlined in the introduction. An important feature of our expansion procedure is the reproducible enrichment in PDX1$^+$/SOX9$^+$/NKX6-1$^+$, which can reach 90%, irrespective of the cell line used. Thus, this procedure will be particularly useful for hPS cell lines with reduced initial capacity to differentiate into PP cells. The comparison of the transcriptome profiles showed that our procedure is unique in effecting a strong upregulation of *GP2*, a unique marker of human fetal PP cells (*Cogger et al., 2017*; *Ramond et al., 2017*) associated with a high PP preference for endocrine differentiation (*Aghazadeh et al., 2022*; *Ameri et al., 2017*). Another unique feature was the strong upregulation of *NKX6-2*, another marker of PP cells that complements *NKX6-1* function but is not retained in differentiated endocrine cells (*Binot et al., 2010*; *Henseleit et al., 2005*; *Nelson et al., 2007*; *Pedersen et al., 2005*; *Schaffer et al., 2010*). The FN procedure appeared to maintain higher levels of TF genes engaged in the duct program, as well as higher levels of *AFP* and *HHEX*, markers of the liver lineage. Expression of acinar *TFs* was very low in all three procedures, and, as expected, the expression of *TFs* driving the endocrine program was strongly reduced in all three procedures albeit our procedure appeared more efficient in this respect, particularly concerning *RFX3* and *RFX6* expression. It is important to point out that all three procedures resulted in noticeable upregulation of *CDX2*, a gut TF, and future efforts should be directed at addressing this, using modifications of the current procedure and FC quantification of CDX2$^+$ cells. Interestingly, all ePP cells clustered separately from human fetal progenitors. This could reflect genuine differences due to culture conditions or differences in developmental age, including the possibility that ePP cells may represent a transient, unstable state in vivo. Another possibility is that PP cells in vivo might be more heterogeneous as it is expected that progenitors at different positions in the developing organ would be at different developmental stages.

Importantly, ePP cells from H1, H9, and CRTD1 hPS cells all differentiated with similar efficiency to dPP cells into SC-islets containing similar numbers of β-cells of comparable functionality. This may appear surprising because ePP cells contained a much higher percentage of PDX1$^+$/SOX9$^+$/NKX6-1$^+$ cells. It should be noted, however, that the liver marker AFP and, particularly, the gut marker CDX2 were upregulated, whereas expression of *PTF1A*, recently shown as essential to set in motion an epigenetic cascade that is required for subsequent duct and endocrine differentiation (*Miguel-Escalada et al., 2022*), was essentially lost. The same pattern, and even higher upregulation of gut and liver markers, was also seen in the other expansion procedures (*Ma et al., 2022*; *Nakamura et al., 2022*).

The unlimited expansion of PP cells reported here is applicable to different hPS cell lines and presents several advantages in the efforts to scale up the generation of islet cells, including β-cells, for the cell therapy of diabetes. It reduces the number of differentiation procedures to be carried out starting at the hPS cell stage, thus eliminating a source of variability and allowing the selection of the most optimally differentiated PP cell population for subsequent expansion and storage. Since it is currently acknowledged that current differentiation procedures do not produce fully functional β-cells, these ePP cells will provide a convenient springboard to refine downstream differentiation procedures. Suitable surface markers for the selection of endocrine cells at the end of the differentiation procedure have been reported (*Li et al., 2020*; *Veres et al., 2019*) but such procedures may prove too expensive in a clinical setting. Alternatively, expanded PP cells could be directly used for transplantations as it has been shown that they can mature in vivo (*Kroon et al., 2008*; *Shapiro et al., 2021*). Whether PP cells or terminally differentiated SC-islet cells would be the best approach in future transplantations is still discussed, but, in any of these cases, the availability of a highly pure GMP-grade, hPS cell-derived PP cell population has several advantages as discussed above.

The transition to full GMP conditions is expected to be relatively straightforward. Human iPS cell lines are already being generated under GMP conditions, and, depending on the culture medium, the first key differentiation step (monolayer hPS cell to definitive endoderm) might need adjustment regarding seeding density, the length of time that hPS cells remain as a monolayer, confluency at the time of differentiation initiation, and possibly concentrations of signals, particularly Activin A and CHIR. In our hands, the plate coating, either with Matrigel or GMP-compatible reagents such as FN or VTN-N, does not have any impact on the differentiation efficiency to definitive endoderm and subsequent stages. Otherwise, the large majority of supplements, used in hPS cell culture and differentiation into SC-islets, are small molecules and recombinant proteins that can be already purchased as GMP reagents. Some small molecules are not yet available as GMP versions, but they could be manufactured upon custom request if necessary.

In summary, the expansion of PP cells will facilitate the generation of an unlimited number of endocrine cells, initially for studying diabetes, drug screening for personalized medicine, and eventually cell therapies. The chemically defined expansion procedure we report here will be an important step toward generating large numbers of human pancreatic endocrine cells that are of great interest for biomedical research and regenerative medicine.

## Materials and methods
### Derivation of the CRTD1 human iPS cell line

The CRTD1 human iPS cell line (hPSCreg: CRTDi004-A) was generated from previously published foreskin fibroblasts (termed Theo) of a consenting healthy donor (*Wolf et al., 2016*). Isolation of cells and reprogramming to hiPS cells was approved by the ethics council of TU Dresden (EK169052010 und EK386102017). Theo fibroblasts were reprogrammed at the CRTD Stem Cell Engineering Facility at the Technical University of Dresden using the CytoTune-iPS 2.0 Sendai Reprogramming Kit (Thermo Fisher Scientific A16517) according to the supplier's recommendations for transduction. Following transduction with the Sendai virus, cells were cultured on irradiated CF1 Mouse Embryonic Fibroblasts (Thermo Fisher Scientific 15943412) in KOSR-based medium containing 80% DMEM/F12 (Thermo Fisher Scientific 31330095), 20% KnockOut Serum Replacement (Thermo Fisher Scientific 10828028), 2 mM L-glutamine (Thermo Fisher Scientific 25030024), 1% nonessential amino acids (Thermo Fisher Scientific 11140035), and 0.1 mM 2-mercaptoethanol (Thermo Fisher Scientific 21985023) supplemented with 10 ng/ml human FGF2 (STEMCELL Technologies 78003). Individual iPSC colonies were mechanically picked, expanded as clonal lines, and adapted to Matrigel (Corning 354277), mTeSR1 (STEMCELL Technologies 85850), and ReLeSR (STEMCELL Technologies 05873) conditions after several passages. Master and working hiPS cell banks were established from clones with the best morphology.

To characterize the newly generated CRTD1 hiPS cell line, several tests were performed, accessible at https://hpscreg.eu/cell-line/CRTDi004-A. Pluripotency was initially analyzed by Alexa Fluor 488-conjugated anti-Oct3/4 (BD Pharmingen 560253), PE-conjugated anti-Sox2 (BD Pharmingen 560291), V450-conjugated anti-SSEA4 (BD Pharmingen 561156), and Alexa Fluor 647-conjugated anti-Tra-1-60 (BD Pharmingen 560122) used according to the manufacturer's recommendations and analyzed on a BD LSRII Flow Cytometer. Then, the three-germ layer differentiation assay was performed as described previously (*Cheung et al., 2011*) and resulting cells were stained using the three-germ layer Immunocytochemistry Kit (Thermo Fisher Scientific A25538) according to the instruction manual. For endoderm differentiation, a SOX17 primary antibody (Abcam ab84990) followed by Alexa Fluor 488 goat anti-mouse IgG (Thermo Fisher Scientific A11001) was used. Quantitative RT-PCR for pluripotency and trilineage spontaneous differentiation was performed according to the instruction manual of the human ES cell Primer Array (Takara Clontech).

Cells were analyzed for chromosomal abnormalities using standard G banding karyotyping. Cells were treated with 100 ng/ml KaryoMAX Colcemid solution (Thermo Fisher Scientific 15212012) for 4 hr at 37°C, harvested and enlarged with 0.075 M KCl solution (Thermo Fisher Scientific 10575090) for 20 min at 37°C. After fixing with 3:1 methanol (VWR 20846.326):glacial acetic acid (VWR 20102.292), cells were spread onto glass slides and stained with Giemsa at the Institute of Human Genetics, Jena University, Germany. G-bandings of at least 20 metaphases were analyzed.

## Isolation of human pancreatic islets

Human islets were obtained through the TU Dresden Islet Transplantation Program approved by the TU Dresden Institutional Review Board (EK 255062022) with written informed consent obtained from each islet donor participant. Islets were isolated and purified from resected pancreas tissue according to a modified Ricordi method. Briefly, collagenase, neutral protease (Serva Electrophoresis, Heidelberg, Germany), and Pulmozyme (Roche, Grenzach, Germany) were infused into the main pancreatic duct. Islets were separated from exocrine tissue by centrifugation on a continuous Biocoll gradient (Biochrom AG, Berlin, Germany) in a COBE 2991 cell processor (Lakewood, CO).

## Maintenance and karyotyping of human pluripotent stem cell lines

The H1 and H9 hES cell lines were purchased from WiCell (Wisconsin, USA). H1 and H9 hES cells as well as CRTD1 iPS cells were maintained on cell culture dishes coated with hES cell qualified Corning Matrigel (BD Biosciences, 354277) diluted 1:50 with DMEM/F-12 (Gibco, 21331-020) and daily changes of mTeSR1 medium (STEMCELL Technologies, 85850) supplemented with 1× penicillin/streptomycin (Gibco, 15140-122). The cells were passaged at around 70% confluency, approximately every 4 d at a ratio of 1:6 to 1:9, as small aggregates using ReLeSR (STEMCELL Technologies, 05872). Karyotyping for H1 and H9 cells was as described above for the CRTD1 iPS cell line, and cells were routinely tested for mycoplasma contamination by PCR as published previously (*Young et al., 2010*) and verified negative.

## Differentiation of hPS cells to PP cells

Initially, the H1 ES cell line was differentiated to PP cells using the STEMdiff Pancreatic Progenitor Kit (STEMCELL Technologies, 05120) according to the manufacturer's instructions. In short, the hES cell colonies were dissociated into single cells using TrypLE Express (Gibco, 12604-013) and seeded on Matrigel-coated plates as described above at a concentration of 95,000 cells/cm$^2$ in mTeSR supplemented with 20 µM ROCKi. The medium was replaced the next day by mTeSR and differentiation was initiated by replacing it with the S1d1 differentiation medium when cells were 60–70% confluent, typically 2 d after the initial seeding. Daily washes with DPBS (Gibco, 14190250) and media changes were done until S4d5 when the cells reached the end of PP stage. The monolayer of PP cells was then dissociated using Accumax (STEMCELL Technologies, 07921), and cells were used for expansion under INI conditions.

Later, H1, H9, and CRTD1 iPS cells were differentiated into PP cells using an adaptation of published procedures (*Mahaddalkar et al., 2020*; *Rezania et al., 2014*; *Shi et al., 2017*; *Supplementary file 1d*). The monolayer of PP cells was then dissociated using TrypLE Express (Gibco, 12604-013) for 2 min, and cells were used for expansion under C0 to C8 conditions.

## Expansion and cryopreservation of PP cells

The monolayer of PP cells was dissociated using TrypLE Express (Gibco, 12604-013), and cells were used for expansion. Expansion cultures were maintained on polystyrene cell culture plates (Corning, CLS3516) coated with Matrigel (Corning, 354277) or Cultrex (R&D Systems, 3434-005-02), both diluted 1:50 in DMEM/F-12 or with 20 ug/mL recombinant truncated vitronectin (VTN-N) (Thermo Fisher Scientific, A31804) diluted in DMEM/F-12 (Gibco, 21331-020) for 1 hr at room temperature (RT). PP cells were resuspended in PP expansion media CINI-C8 (*Supplementary file 1a*) and seeded initially at a density of $3.2 \times 10^5$/ cm$^2$. In subsequent passages, cells were seeded at a density of $2.1 \times 10^5$/cm$^2$. In every passage, expansion media were supplemented, during the first day, with 10 µM ROCKi. The expansion medium was changed daily, and cells were typically passaged every fourth day using TrypLE Express dissociation into single cells. Karyotyping for expanded PP cells and mycoplasma testing was as described above for the hPS cell lines.

Expanding PP cells were routinely frozen at later passages using mFreSR (STEMCELL Technologies, 05854), supplemented with 20 µM ROCKi, at a density of 10 million cells/ml. To ensure proper controlled freezing (−1°C/min), cryotubes were placed in Mr. Frosty Freezing Container (Thermo Fisher Scientific, 5100-0001) at −80°C. After 24 hr, cryotubes were transferred to a liquid nitrogen chamber. For thawing, frozen cells were placed at 37°C and then transferred to 6 ml DMEM/F-12 at RT for centrifugation. After spinning down the cells at 600 × *g*, the pellet was resuspended using PP expansion media and cells were counted using the Countess II Automated Cell Counter (Thermo

Fisher Scientific) and Trypan blue. Typical recovery rates were above 85%. Cells were then seeded as described above for expansion.

## Differentiation of CINI ePP cells into pancreatic endocrine cells using ALI culture

PP cells generated using the STEMdiff Pancreatic Progenitor Kit and expanded under CINI conditions were dissociated into single cells using Accutase (STEMCELL Technologies, 07920) at 37°C and then resuspended in PEP medium (*Supplementary file 1d*) supplemented with 10 μM ROCKi at a concentration of 50,000 cells/μl. The Falcon Cell Culture Inserts (Corning, 353493) were placed in their companion plate wells (Corning, 353502) containing 1.5 ml of complete PEP media supplemented with 10 μM ROCKi. Ten droplets of 5 μl of the cell suspension were dropped on the insert to create 10 3D clusters per well, each containing 250,000 cells. Daily media changes were conducted according to *Supplementary file 1d* with the following changes: (a) during S5, the XX Notch inhibitor (Millipore, 565789) was added at a concentration of 10 nM; (b) final glucose concentration at S6 and S7 was 20 mM; and (c) no sodium pyruvate was used at S7.

## Differentiation of PP cells into SC-islets using microwells

PP cells were dissociated using TrypLE Express at 37°C for 2–3 min and seeded in microwells of AggreWell 800 plates (STEMCELL Technologies, 34825) using the recommended procedure by the supplier. Directly differentiated and ePP cells were seeded at a density of 5000 or 2000 cells per microwell, respectively. Media were an adaptation of published procedures (*Balboa et al., 2022*; *Mahaddalkar et al., 2020*; *Rezania et al., 2014*; *Shi et al., 2017*; *Supplementary file 1d*). Expanded PP cells in micropatterned wells were kept in expansion media for 24 hr, and, during the first day of S5, S5 medium was supplemented with one-fourth of the concentration of C6 additional factors. Media of S5–S7 are as described in *Supplementary file 1d*, and the medium was changed daily.

## Immunofluorescence analyses

For immunofluorescence (IF), cells were cultured on 12-mm-diameter Matrigel-coated coverslips (Carl Roth, P231.1) placed in 12-well wells (Corning, CLS3513) and differentiated or expanded as described above. Cells were then fixed in 4% paraformaldehyde (PFA) for 20 min at 4°C and washed with PBS. Cells were blocked and permeabilized for 1 hr at RT using a 5% serum/0.3% Triton X-100 PBS solution. Samples were then incubated at 4°C in 2.5% serum/0.3% Triton X-100 in PBS containing the primary antibodies in the appropriate concentration (*Supplementary file 1f*). The following day, samples were incubated for 1 hr at RT in 2.5% serum/0.3% Triton X-100 in PBS containing the appropriate secondary antibodies, conjugated with either Alexa 488, 568, or 647, at a 1:500 dilution (*Supplementary file 1g*). The coverslip with the stained cells was then placed on a microscope slide, covered with ProLong Gold Antifade mounting medium with DAPI (Invitrogen, P36931), and overlayed with a rectangular coverslip.

IF images were acquired using a Zeiss Axio Observer Z1 microscope coupled with the Apotome 2.0 imaging system and with consistent exposure times for the Alexa 488, 568m and 647 channels in between passages and conditions to allow for direct comparison of the signal intensities.

## Fluorescence-activated cell sorting (FACS) of hPS cell-derived PP and endocrine cells

Cells were first washed with 2% BSA in DMEM-F12 and then dissociated into single cells using TrypLE Express (Gibco, 12604-013). Following dissociation, the cells were counted using the Countess II Automated Cell Counter (Thermo Fischer Scientific) and Trypan blue. Then, cells were washed with PBS and fixed at a concentration of 4 million/ml in 4% PFA for 10 min at 4°C. For staining with TF antibodies, cells were washed with PBS and permeabilized using the Foxp3 Transcription Factor Staining Buffer Set (Invitrogen, 00-5523-00) for 1 hr in dark at 4°C. Cells were then washed again using the 1× permeabilization buffer. Thereafter, $2 \times 10^6$ cells/sample were blocked using 100 μl of 5% serum in 1× permeabilization buffer and then incubated with primary antibodies at the appropriate concentration (*Supplementary file 1e*) in the same buffer overnight at 4°C. Then, they were washed twice with 1× permeabilization buffer and incubated with secondary antibodies at the appropriate concentration (*Supplementary file 1g*) in the same buffer at RT for 1 hr in dark. For cytoplasmic factors, $2 \times 10^6$ cells/

sample were blocked for 30 min at RT in 100 µl PBS containing 2% serum and 0.3% Triton X-100. Then cells were washed with 0.1% Triton X-100 in PBS solution and incubated with primary antibodies at the appropriate concentrations (*Supplementary file 1e*) in the same buffer overnight. Thereafter, cells were washed twice with 0.1% Triton X-100 in PBS solution and incubated with secondary antibodies at the appropriate concentration (*Supplementary file 1g*) at RT in the dark for 1 hr. For conjugated antibodies, the appropriate number of cells was used, as directed by the manufacturer, for both the isotype control and staining sample. Samples were stained overnight with conjugated antibodies and then washed with 0.1% Triton X-100 in PBS solution. FACS data were acquired using BD FACS Canto II and analyzed using the FlowJo software.

## Proliferation and cell death assays

The EdU proliferation assay was performed with the Click-iT Plus EdU Alexa Fluor 488 Flow Cytometry Assay Kit (Invitrogen, C10632), which contains all the necessary reagent except PBS and BSA, and according to the kit protocol. In short, on the day of passaging the expanding cells, the expansion media was supplemented with 10 µM EdU for 2 hr and cells were then dissociated and washed in 3 ml of 1% BSA in PBS. A pellet of 1 million cells was resuspended in 100 µl of Click-iT fixative for 15 min and washed again with 3 ml of 1% BSA in PBS. The pelleted cells were resuspended in 100 µl of permeabilization and wash reagent for 15 min. Following the incubation, 0.5 ml of the Click-iT Plus reaction cocktail containing 1× buffer additive, the Alexa Fluor 488 picolyl azide and a copper protectant in PBS, was added to the sample and incubated in the dark at RT for 30 min. The cells were then washed with 3 ml of 1× permeabilization and wash reagent. The pelleted cells were resuspended in 0.5 ml of permeabilization and wash reagent, passed through a 40 µm strainer (PluriSelect, 43-10040), and analyzed on the BD FACS Canto II.

The flow cytometry staining for necrosis and apoptosis was performed with the FITC Annexin V Apoptosis Detection Kit with 7-AAD (BioLegend, 640922). Once the cells were incubated in 10 µM of EdU, they were detached and washed in 1% BSA in PBS and an aliquot of 250,000 cells was taken and pelleted in a 1.5 ml microcentrifugation tube. The cell pellet was resuspended in 100 µl of Annexin V Binding Buffer containing 5 µl of FITC Annexin V and 5 µl of 7-AAD Viability Staining Solution, and cells were incubated for 15 min at RT in the dark. After the incubation, an additional 400 µl of Annexin V Binding Buffer was added to the sample and passed through a 40 µm strainer (PluriSelect, 43-10040) before the live cells were analyzed on the BD FACS Canto II.

## Static glucose-stimulated insulin secretion (GSIS) assay

At the end of S7 (S7d10–S7d14), 150 clusters were collected, washed with PBS, and incubated for 1 hr in 700 µl of fresh Kreb's buffer (2.5 mM CaCl$_2$, 129 mM NaCl, 4.8 mM KCl, 1.2 mM MgSO$_4$, 1.2 mM KH$_2$PO$_4$, 1 mM Na$_2$HPO$_4$, 5 mM NaHCO$_3$, 10 mM HEPES, 0.1% BS, pH adjusted to 7.4 with 5 M NaOH). After first incubation, clusters were incubated with Kreb's buffer containing low glucose (2.8 mM) for 1 hr, then high glucose (16.7 mM) for 1 hr, and finally KCl/high glucose (30 mM/16.7 mM) for 1 hr. Incubations were for exactly 1 hr, and then supernatant was collected. Collected supernatants and pelleted cells were frozen at –80°C until the analysis. C-peptide detection was performed using the human C-peptide ELISA kit (Mercodia, 10-1141-01), readings were taken using an ELISA plate reader, and the standard curve was generated. Cell pellets were then used to isolate genomic DNA with the DNeasy Blood & Tissue kit (QIAGEN, 69504), and DNA quantification was done using a NanoDrop spectrophotometer.

## RNA isolation and qRT-PCR (qPCR)

Total RNA was prepared using the RNeasy kit with on-column genomic DNA digestion (QIAGEN, 74004) following the manufacturer's instructions. First-strand cDNA was prepared using the TAKARA PrimeScript RT Master Mix (TAKARA RR036A). Real-time PCR primers (*Supplementary file 1h*) were designed using the Primer 3 software (SimGene), their specificity was ensured by in silico PCR, and they were further evaluated by inspection of the dissociation curve. Reactions were performed with the FastStart Essential DNA Green Master mix (Roche 06924204001) using the Roche LightCycler 480, and primary results were analyzed using the on-board software. Reactions were carried out in technical triplicates from at least three independent biological samples. Relative expression values

were calculated using the ΔΔCt method by normalizing to H1 undifferentiated expression levels and the *TBP* housekeeping gene.

## RNA sequencing and bioinformatics analysis

Cells were differentiated and from hPS cells as described above. Three independent samples from distinct differentiations and independent expansions were used as biological replicates. Total RNA prepared as above with an integrity number of ≥9 was used, and subsequent steps were performed at the Biotec Sequencing Core of TU Dresden. mRNA was isolated from 1 ug of total RNA by poly-dT enrichment using the NEBNext Poly(A) mRNA Magnetic Isolation Module according to the manufacturer's instructions. Final elution was done in 15 ul 2× first-strand cDNA synthesis buffer (NEBnext, NEB). After chemical fragmentation by incubating for 15 min at 94°C, the sample was directly subjected to the workflow for strand-specific RNA-Seq library preparation (Ultra Directional RNA Library Prep, NEB). For ligation, custom adaptors were used 1: (Adaptor-Oligo 5'-ACA CTC TTT CCC TAC ACG ACG CTC TTC CGA TCT-3', Adaptor-Oligo 2: 5'-P-GAT CGG AAG AGC ACA CGT CTG AAC TCC AGT CAC-3'). After ligation, adapters were depleted by an XP bead purification (Beckman Coulter) adding bead in a ratio of 1:1. Indexing was done during the following PCR enrichment (15 cycles) using custom amplification primers carrying the index sequence indicated with 'NNNNNN'. Primer 1: AAT GAT ACG GCG ACC ACC GAG ATC TAC ACT CTT TCC CTA CAC GAC GCT CTT CCG ATC T; primer 2: GTG ACT GGA GTT CAG ACG TGT GCT CTT CCG ATC T; primer 3: CAA GCA GAA GAC GGC ATA CGA GAT NNNNNN GTG ACT GGA GTT. After two more XP beads purifications (1:1), libraries were quantified using Qubit dsDNA HS Assay Kit (Invitrogen). For Illumina flow cell production, samples were equimolarly pooled and distributed on all lanes used for 75 bp single-read sequencing on Illumina HiSeq 2500.

After sequencing, FastQC (http://www.bioinformatics.babraham.ac.uk/) was used to perform a basic quality control of the resulting sequencing data. Fragments were aligned to the human reference genome hg38 with support of the Ensembl 104 splice sites using the aligner gsnap (v2020-12-16) (*Wu and Nacu, 2010*). Counts per gene and sample were obtained based on the overlap of the uniquely mapped reads with the same Ensembl annotation using featureCounts (v2.0.1) (*Liao et al., 2014*). Normalization of raw fragments based on library size and testing for differential expression between the different cell types/treatments was done with the DESeq2 R package (v1.30.1) (*Love et al., 2014*). Sample-to-sample Euclidean distance, Pearson and Spearman correlation coefficient (r), and PCA based upon the top 500 genes showing highest variance were computed to explore the correlation between biological replicates and different libraries. To identify DEGs, counts were fitted to the negative binomial distribution and genes were tested between conditions using the Wald test of DESeq2. Resulting p-values were corrected for multiple testing with the using Independent Hypothesis Weighting (v1.18.0) (*Ignatiadis et al., 2016*). Genes with a maximum of 5% false discovery rate ($p_{adj}$≤0.05), 0.5 > fold regulation > 2.0, and counts above 200 were considered as significantly differentially expressed.

To directly compare the transcriptome profile of our expanded PP cells with previously published datasets, raw sequencing data of the GEO archives GSE156712 (*Ma et al., 2022*) were downloaded. The array express, E-MTAB-9992 archive from EBI (*Nakamura et al., 2022*) and EGAS00001003127 from EGA (*Ramond et al., 2018*), offered bam files. Here, the fastq files were extracted with picard tools (v2.25.6). All these datasets underwent the same processing procedure. Fragments/reads were aligned to the human reference genome hg38 with support of the Ensembl 104 splice sites using the aligner gsnap (v2020-12-16). Counts per gene and sample were obtained based on the overlap of the uniquely mapped reads with the same Ensembl annotation using featureCounts (v2.0.1) (*Liao et al., 2014*). The various strand-specificity of several projects was taken into account for the gene counting. Normalization of raw fragments based on library size and scaling the count on a log2 scale was done with the DESeq2 R package (v1.30.1) (*Love et al., 2014*) and the Variance Stabilizing Transformation (vst) function. These values were used for plotting.

Original RNA-Seq data have been deposited in GEO under the GSE216179 (CINI ePP transcriptome) and GSE216266 (C6 ePP transcriptome) accession numbers. GO and KEGG analyses have been carried out using the Enrichr suite (https://maayanlab.cloud/Enrichr/ ; *Kuleshov et al., 2016*) and GSEA using the UCSD Broad Institute suite (https://www.gsea-msigdb.org/gsea/index.jsp; *Kuleshov et al., 2016*). Heat maps were generated using the Morpheus application (https://software.broadinstitute.org/morpheus/).

## Acknowledgements

Research in the AG laboratory was supported by grants from the German Center for Diabetes Research (DZD) (grant 82DZD00101) and the German Research Foundation (DFG) (grants GA-2004/3-1 and IRTG 2251). Fibroblasts 'Theo', used for the generation of CRTD1 hiPSC, were a gift from Prof. Dr. Med. Min Ae Lee-Kirsch, University Hospital Carl Gustav Carus, Dresden.

## Additional information

### Funding

| Funder | Grant reference number | Author |
|---|---|---|
| Helmholtz Zentrum München | 82DZD00101 | Anthony Gavalas |
| Deutsche Forschungsgemeinschaft | GA-2004/3-1 | Anthony Gavalas |
| Deutsche Forschungsgemeinschaft | IRTG 2251 | Anthony Gavalas |

The funders had no role in study design, data collection and interpretation, or the decision to submit the work for publication.

### Author contributions

Luka Jarc, Manuj Bandral, Data curation, Formal analysis, Investigation, Methodology, Writing – review and editing; Elisa Zanfrini, Investigation, Methodology; Mathias Lesche, Data curation, Formal analysis; Vida Kufrin, Raquel Sendra, Daniela Pezzolla, Ioannis Giannios, Investigation; Shahryar Khattak, Barbara Ludwig, Resources; Katrin Neumann, Resources, Methodology; Anthony Gavalas, Conceptualization, Data curation, Formal analysis, Supervision, Funding acquisition, Writing - original draft, Project administration, Writing – review and editing

### Author ORCIDs

Vida Kufrin ⬡ http://orcid.org/0009-0002-2431-7654
Raquel Sendra ⬡ http://orcid.org/0000-0001-7158-4582
Anthony Gavalas ⬡ http://orcid.org/0000-0003-4897-0359

### Ethics

Human subjects: Human islets were obtained through the TU Dresden Islet Transplantation Program approved by the TU Dresden Institutional Review Board (EK 255062022) with written informed consent obtained from each islet donor participant.

Reviewer #1 (Public Review): https://doi.org/10.7554/eLife.89962.3.sa1
Reviewer #2 (Public Review): https://doi.org/10.7554/eLife.89962.3.sa2
Author Response https://doi.org/10.7554/eLife.89962.3.sa3

## Additional files

### Supplementary files

• Supplementary file 1. Tables. (**a**) Expansion conditions tested and corresponding doubling times. ND; not determined. (**b**) Regulated genes in CINI expansion. In group 1, there are genes continuously upregulated in p5 and p10, in group 2 genes upregulated at p5 but stable at p10, in group 3 genes downregulated at p5 but stable in p10, and in group 4 genes continuously downregulated. (**c**) Gene regulation of components of selected signaling pathways during expansion in CINI. (**d**) Basal media and supplements for S1–S7. (**e**) Normalized RNA-Seq counts of selected genes from feeder-expanded cells, fibronectin-expanded cells, and vitronectin-N-expanded cells. (**f**) List of primary antibodies for immunofluorescense (IF) and flow cytometry (FC).

• MDAR checklist

## Data availability

Original RNA Seq data have been deposited in GEO under the GSE216179 (CINI ePP transcriptome) and GSE216266 (C6 ePP transcriptome) accession numbers.

The following datasets were generated:

| Author(s) | Year | Dataset title | Dataset URL | Database and Identifier |
|---|---|---|---|---|
| Giannios I, Pezzolla D, Gavalas A | 2023 | Transcriptome profiling of expanding pancreas progenitors derived from the H1 human pluripotent stem cells using TGFb inhibition | https://www.ncbi.nlm.nih.gov/geo/query/acc.cgi?acc=GSE216179 | NCBI Gene Expression Omnibus, GSE216179 |
| Bandral M, Jarc L, Gavalas A | 2023 | Transcriptome profiling of expanding pancreas progenitors derived from the H1 human pluripotent stem cells using a combination of mitogenic signals, RA signalling suppression and selective inhibition of the TGFb and Wnt signalling pathways | https://www.ncbi.nlm.nih.gov/geo/query/acc.cgi?acc=GSE216266 | NCBI Gene Expression Omnibus, GSE216266 |

The following previously published datasets were used:

| Author(s) | Year | Dataset title | Dataset URL | Database and Identifier |
|---|---|---|---|---|
| Ma X, Lu Y, Zhou Z Li Q | 2020 | Robust Expansion of Human Pancreatic Progenitors by Small Molecules | https://www.ncbi.nlm.nih.gov/geo/query/acc.cgi?acc=GSE156712 | NCBI Gene Expression Omnibus, GSE156712 |
| Wong YF, Michaut M | 2021 | A long-term feeder-free culture method for human pancreatic progenitors on extra cellular matrix or matrix-free polymers | https://www.ebi.ac.uk/biostudies/arrayexpress/studies/E-MTAB-9992 | ArrayExpress, E-MTAB-9992 |
| Ramond C, Hansson M | 2018 | RNAseq of human fetal pancreas development | https://ega-archive.org/datasets/EGAD00001004210 | European Genome-Phenome Archive, EGAD00001004210 |

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
