## [Editor Report · eLife assessment]

This **important** study describes a method to decouple the mechanisms supporting pancreatic progenitor self-renewal and expansion from feed-forward mechanisms promoting their differentiation allowing in vitro expansion of hPSC-derived pancreatic progenitors. The strength of evidence is **convincing** in that the authors use appropriate and validated methodology in line with current state-of-the-art. The work will be of interest to the field of β-cell replacement therapy in diabetes.

---

## [Referee Report · Reviewer #2 (Public review)]

The paper presents a novel approach to expand iPSC-derived pdx1+/nkx6.1+ pancreas progenitors, making them potentially suitable for GMP-compatible protocols. This advancement represents a significant breakthrough for diabetes cell replacement therapies, as one of the current bottlenecks is the inability of expanding PP without compromising their differentiation potential. The study employs a robust dataset and state-of-the-art methodology, unveiling crucial signaling pathways (eg TGF, Notch...) responsible for sustaining pancreas progenitors while preserving their differentiation potential in vitro.

The current version of the paper has improved, increasing the clarity and providing clear explanations to the comments raised regarding quantifications, functionality of the cells in vivo etc...

The discussion on challenges adds depth to the study and encourages future research to build upon these important findings

---

## [Referee Report · Reviewer #3 (Public review)]

In this work, Jarc et al. describe a method to decouple the mechanisms supporting progenitor self-renewal and expansion from feed-forward mechanisms promoting their differentiation.

The authors aimed at expanding pancreatic progenitor (PP) cells, strictly characterized as PDX1+/SOX9+/NKX6.1+ cells, for several rounds. This required finding the best cell culture conditions that allow sustaining PP cell proliferation along cell passages while avoiding their further differentiation. They achieve this by comparing the transcriptome of PP cells that can be expanded for several passages against the transcriptome of unexpanded (just differentiated) PP cells.

The optimized culture conditions enabled the selection of PDX1+/SOX9+/NKX6.1+ PP cells and their consistent, 2000-fold, expansion over ten passages and 40-45 days. Transcriptome analyses confirmed the stabilization of PP identity and the effective suppression of differentiation. These optimized culture conditions consisted in substituting the Vitamin A containing B27 supplement with a B27 formulation devoid of vitamin A (to avoid retinoic acid (RA) signaling from an autocrine feed-forward loop), substituting A38-01 with the ALK5 II inhibitor (ALK5i II) that targets primarily ALK5, supplementation of medium with FGF18 (in addition to FGF2) and the canonical Wnt inhibitor IWR-1, and cell culture on vitronectin-N (VTN-N) as a substrate instead of Matrigel.

The strength of this work relies on a clever approach to identify cell culture modifications that allow expansion of PP cells (once differentiated) while maintaining, if not reinforcing, PP cell identity. Along the work, it is emphasized that PP cell identity is associated to the co-expression of PDX1, SOX9 and NKX6.1. The optimized protocol is unique (among the other datasets used in the comparison shown here) at inducing a strong upregulation of GP2, a unique marker of human fetal pancreas progenitors. Importantly GP2+ enriched hPS cell-derived PP cells are more efficiently differentiating into pancreatic endocrine cells (Aghazadeh et al., 2022; Ameri et al., 2017).

The unlimited expansion of PP cells reported here would allow scaling-up the generation of beta cells, for the cell therapy of diabetes, by eliminating a source of variability derived from the number of differentiation procedures to be carried out when starting at the hPS cell stage each time. The approach presented here would allow selection of the most optimally differentiated PP cell population for subsequent expansion and storage. Among other conditions optimized, the authors report a role for Vitamin A in activating retinoic acid signaling in an autocrine feed-forward loop, and the supplementation with FGF18 to reinforce FGF2 signaling.

This is a relevant topic in the field of research, and some of the cell culture conditions reported here for PP expansion might have important implications in cell therapy approaches. Thus, the approach and results presented in this study could be of interest for researchers working in the field of in vitro pancreatic beta cell differentiation from hPSCs. Table S1 and Table S4 are clearly detailed and extremely instrumental to this aim.

---

## [Author Response]

The following is the authors’ response to the original reviews.

**eLife assessment**
The authors describe a method to decouple the mechanisms supporting pancreatic progenitor self-renewal and expansion from feed-forward mechanisms promoting their differentiation. The findings are important because they have implications beyond a single subfield. The strength of evidence is solid in that the methods, data and analyses broadly support the claims with only minor weaknesses.

We are grateful for the substantial effort that reviewers put into reading our manuscript and providing such a detailed feedback. We have strived to address, as much as possible, all comments and criticisms. Thanks to the feedback, we believe that we have now a significantly improved manuscript. Below, there is a point-bypoint response.

**Reviewer #1 (Public Review)**
In this manuscript, the authors are developing a new protocol that aims at expanding pancreatic progenitors derived from human pluripotent stem cells under GMP-compliant conditions. The strategy is based on hypothesis-driven experiments that come from knowledge derived from pancreatic developmental biology.The topic is of major interest in the view of the importance of amplifying human pancreatic progenitors (both for fundamental purposes and for future clinical applications). There is indeed currently a major lack of information on efficient conditions to reach this objective, despite major recurrent efforts by the scientific community.Using their approach that combines stimulation of specific mitogenic pathways and inhibition of retinoic acid and specific branches of the TGF-beta and Wnt pathways, the authors claim to be able, in a highly robust and reproducible manner to amplify in 10 passages the number of pancreatic progenitors (PP) by 2,000 folds, which is really an impressive breakthrough.The work is globally well-performed and quite convincing. I have however some technical comments mainly related to the quantification of pancreatic progenitor amplification and to their differentiation into beta-like cells following amplification.

We thank the reviewer for the positive assessment. Below we provide a point-by-point response to specific comments and criticisms.

**Reviewer #1 (Recommendations For The Authors)**
Figure 1:Panel A: What is exactly counted in Fig. 1A? Is it the number of PP (as indicated in the title) or the total number of cells? If it is PPs, was it done following PDX1/NKX6.1/SOX9 staining and FACS quantification? This question applies to a number of Figures and the authors should be clear on this point.

We now define ‘PP cells’ as ‘PP-containing cells’ (PP cells) the first time we use the term in the RESULTS section.

Panel D: I do not understand the source of TGFb1, GDF11, FGF18, PDGFA. Which cell type(s) express such factors in culture? I was not convinced that the signals are produced by PP and act through an autocrine loop. I have the same type of questions for the receptors: PDGFR on the second page of the results; RARs and RXRs on the third page.

We refer to these factors/receptors as components of a tentative autocrine loop. We agree we do not prove it and we now comment on this in the discussion section.

Figure 2:FACS plots are very difficult to analyze for two reasons: I do not understand the meaning of the y axes (PDX1/SOX9). Does that mean that 100% of the cells were PDX1+/SOX9+? The authors should show the separated FACS plots. More importantly, the x axes indicate that NKX6.1 FACS staining is very weak. This is by far different from what can be read in publications performing the same types of experiments (publications by Millman, Otonkoski...as examples). How was quantification performed when it is so difficult to properly define positive vs negative populations? It is necessary to present proper "negative controls" for FACS experiments and to clearly indicate how positive versus cells were defined

We now explain the gating strategy better in the results section, all controls are included in figure S2.

Figure 3:What is the exact "phenotype" of the cells that incorporated EdU: It would be really instructive to addPDX1/NKX6.1/SOX9 staining on top of EdU. I am also surprised that 20% of the cells stain positive for Annexin V. This is a huge fraction. Does that mean that many cells (20%) are dying and if the case, how amplification can take place under such deleterious conditions?

This is an interesting mechanistic point but performing these experiments would delay the publication of the final manuscript for too long. These assays were done at p3 in order to catch CINI cells that do not expand in most cases. It is important to note that cell death also appears higher in CINI cells. It is likely that the combination of these effects results in reproducible expansion under C5. We comment on the possibilities in the discussion section.

Figure 4:On FACS plots the intensity at the single cell level (see x-axis of the figure) of the NKX6.1 staining is found to increase in Fig. 4G by 50-100 folds when compared to Fig. 4E. Is it expected? This should be discussed in the text. Do the authors observe the same increase by immunocytochemistry?

The apparent difference is actually 10-fold (from 2x102 to 2x103). We think that the most likely reason for this apparent increase is that at p0 we typically used very few cells for the FC in order to keep as many as possible for the subsequent expansion. If we had used more, we would be able to also detect cells with higher expression. As we mention in the bioinformatics analysis, NKX6 expression does increase with passaging and therefore it is also possible that at least part of this increase is real. However, we don’t have suitable data (same number of cells analyzed at each passage) to address this in a reliable manner.

Figure 5Previous data from the scientific literature indicate that in vitro, by default, PP gives rise to duct-like cells. This is a bit described in the result section and supplementary figures taking into account the expression of transcription factors. However the data are not clearly explained and described in quite a qualitative manner. They should appear in a quantitative fashion (and the main figures), adding additional duct cell markers such as Carbonic anhydrase, SPP1, CFTR, and others. I assume that the authors can easily use their transcriptomic data to produce a Figure to be described and discussed in detail.

We think it can be misleading to use such markers (other than TFs and the latter only as a collective) because specific markers of terminal differentiation are more often than not expressed during development in multipotent progenitors, the most conspicuous example been CPA1. To illustrate the point, we used the RNASeq data of and plotted the expression values of a panel of duct genes in isolated human fetal progenitors (Ramond et al., 2017) together with their expression in p0 PP and ePP cells from all three different procedure (please see below). All raw RNA Seq data were processed together to enable direct comparison. According to the analysis of Ramond et al the A population corresponds to MPCs, C to early endocrine progenitors (EP), D to late endocrine progenitors and, by inference and gene expression pattern B to BPs. Expression levels of all these markers were very similar suggesting that these markers cannot be used to distinguish between duct cells and progenitor cells. Importantly, SC-islets derived from either dPP or ePP cells express extremely low and similar levels of KRT19, a marker of duct cells. This latter information is now included in the last part of the results (Figure S7).

Fig. 7:The figure is a bit disappointing for 2 reasons. In A and B, the quality of INS, GCG, and SST staining is really poor. In E, GSIS is really difficult to interpret. They should not be presented as stimulatory indexes. The authors should present independently: INS content; INS secretion at low glucose; INS secretion at high glucose; INS secretion with KCL. Finally, the authors should indicate that glucose poorly (around 2 fold) activates insulin/C-Pept secretion in their stem-cell-derived islets.

We disagree with the quality assessment of the immunofluorescence. Stimulation indexes are also used very widely but we now provide data for actual C-peptide secretion normalized for DNA content of the SC-islets. For technical reasons we do not have normalized C-peptide secretion for human islets. However, we provide a direct comparison to the stimulation index of human islets assayed under the same conditions (2.7 mM glucose / 16.7 mM glucose / 16.7 mM glucose + 30 mM KCl) without presenting SC-islets separately and tweaking the glucose basal (lowering) and stimulation (increasing) levels to inflate the stimulation index. This is unfortunately common. In any case, we do not claim an improvement in the differentiation conditions and our S5-S7 steps may not be optimal but this is not the subject of this work.

**Reviewer #2 (Public Review)**
SummaryThe paper presents a novel approach to expand iPSC-derived pdx1+/nkx6.1+ pancreas progenitors, making them potentially suitable for GMP-compatible protocols. This advancement represents a significant breakthrough for diabetes cell replacement therapies, as one of the current bottlenecks is the inability to expand PP without compromising their differentiation potential. The study employs a robust dataset and state-of-the-art methodology, unveiling crucial signaling pathways (eg TGF, Notch...) responsible for sustaining pancreas progenitors while preserving their differentiation potential in vitro.StrengthsThis paper has strong data, guided omics technology, clear aims, applicability to current protocols, and beneficial implications for diabetes research. The discussion on challenges adds depth to the study and encourages future research to build upon these important findings.

We thank the reviewer for the positive assessment. Below we provide a point-by-point response to general comments and criticisms.

WeaknessesThe paper does have some weaknesses that could be addressed to improve its overall clarity and impact. The writing style could benefit from simplification, as certain sections are explained in a convoluted manner and difficult to follow, in some instances, redundancy is evident. Furthermore, the legends accompanying figures should be self-explanatory, ensuring that readers can easily understand the presented data without the need to be checking along the paper for information.

We have simplified the text in several places and removed redundancies, particularly in the discussion. We revisited the figure legends and made minor corrections to increase clarity. However, regarding the figure legends, we think that adding the interpretation of the results would be redundant to the main text.

The culture conditions employed in the study might benefit from more systematic organization and documentation, making them easier to follow.There is a comparative Table (Table S1) where all conditions are summarized. We refer to this Table every time that we introduce a new condition. We also have a Table (Table S4) which presents all different media and components used it the differentiation procedure.Another important aspect is the functionality of the expanded cells after differentiation. While the study provides valuable insights into the expansion of pancreas progenitors in vitro and does the basic tests to measure their functionality after differentiation the paper could be strengthened by exploring the behavior and efficacy of these cells deeper, and in an in vivo setting.

This will be done in a future study where we will also introduce a number of modifications in S5-S7

Quantifications for immunofluorescence (IF) data should be displayed.

We have not conducted quantifications of IFs because FC is much more objective and accurate. We have not conducted FC for CDX2 and AFP because all other data strongly favor C6 anyway. It should be noted that CDX2 and AFP expression is generally not addressed at all presumably because it raises uncomfortable questions and, to our knowledge, we are the first to address this so exhaustively.

Some claims made in the paper may come across as somewhat speculative.

We have now indicated so where applicable.

Additionally, while the paper discusses the potential adaptability of the method to GMP-compatible protocols, there is limited elaboration on how this transition would occur practically or any discussion of the challenges it might entail.

We have now added a paragraph discussing this in the discussion section.

**Reviewer #2 (Recommendations For The Authors)**
Related to Figure 1:Unclear if CINI or SB431542 + CINI was used (first paragraph of results...)

The paragraph was unclear and it is now rewritten

Was the differentiation to PP similar between the different attempts? A basic QC for each Stem Cell technology differentiation would be good to include.

We added (Figure 1B) a comparison of expression data of general genes (QC) in PP cells showing very comparable patterns of expression. Some of these PP cells went on to expand and most did not but there is no apparent correlation of this with the gene expression data.

qPCR data - relative fold? over what condition? (indicate on axis label)

We added a label as well as an explanation on p0 values in the figure legend

FGF18/ PDGFA - worth including background in pancreas development as in the other factors.

Background information has been added

Bioinformatics is a bit biased with a few genes selected - what are the DEGs / top enriched pathways? Maybe worth showing a volcano plot of the DEGs for example.

We have done all these standard analyses but we think that they did not contribute anything else useful to the study with the exception of pointing to the finding that the TGFb pathway is negatively correlated with expansion, and this is included in the study. The ‘unbiased’ analysis that the reviewer suggests did not turn out something else useful to exploit for the expansion. This does not mean that our approach is biased – in our view it is hypothesis-driven. As we also write in the manuscript, if in a certain pathway a key gene fails to be expressed, the pathway will not show up in any GO or GSEA analyses. However, the pathway will still be regulated. The RA and FGF18 cases clearly illustrate this. We realize that these analyses have become a standard but we think that it is not the only way to approach genomics data and these approaches did not offer much in the context of this study.

The E2F part is very speculative

The pathway came up as a result of ‘unbiased’ GSEA analyses. However, we do agree and rephrased.

The authors claim ' the negative correlation of TGFb signalling with expansion retrospectively justifies the use of A83 '. However, p0 is not treated with A83 - how can they tell that there is a correlation between TGFb signalling and expansion?

The correlation came from the RNA Seq data analysis during expansion. We have rephrased slightly to convey the message more clearly.

Typo with TGFbeta inhibitor name is mispelled (A3801)

Corrected

Page 5 - last paragraph - Table S3? (isnt it refering to S2?)

Since Table S2 is the list of the regulated genes and S3 is the list of the regulated signaling pathway components both are relevant here, we now refer to both.

In the text Figure 2G should read Figure 1G (page 7, end of 1st paragraph).

Corrected

'Autocrine loop' existence – speculative

Added the phrase ‘we speculated’. We refer to this only as a tentative interpretation. We also elaborate in the discussion now.

Related to Figure 2:I am not sure if I would refer to chemical "activation/inhibition" of pathways as 'gain/loss of function'. Maybe this term is more adequate for genetic modifications.

For genetic manipulations, these terms are (supposed to be) accompanied by the adjective ‘genetic’ but to avoid misinterpretations we changed the terms to activation and inhibition as suggested.

It would be good to include a summary of the different conditions as a schematic in one of the figures, to make it very clear to the reader what the conditions are.

We tried this in an early version of the manuscript but, in our view, it was adding complexity, rather than simplifying things. The problem is that as such the Table cannot be integrated in any figure if eg in Figure 2 it would be too early, if in Figure 4 it would be too late and so on. All conditions show up in detail in Table S1.

Nkx6.1 - is the image representative? It looks like Nkx6.1 decreases over the passages.

We do mention in the text that ‘… even though expansion (in C5) appeared to somewhat reduce the number of NKX6.1+ cells. (Figure 2E-G). As we mentioned, this was one of the reasons to continue with other conditions (C6-C8).

Upregulation of AFP/ CDX2 is a bit concerning - the IF for C5 p5 shows a high proportion of CDX2+ cells (Fig S2I). perhaps it would be good to quantify the IF.

It was concerning – this is why we then tested conditions C6-8. Since it is C6 that we propose at the end, it would be, in our view, extraneous to quantify CDX2 in C5.

How do C5/C1/C0 compare to CINI?

We now remind the reader in the results section that CINI was not reproducible - so any other comparison would be extraneous.

Related to Figure 3:There is a 'Lore Ipsum' label above B

Corrected

Related to Figure 4:It is good that AFP expression is reduced at p10, but there seems to be a high proportion of AFP at p5. IF/FACS should be quantified.

We think that this would not add significantly since there are several other criteria, particularly the increase of the PDX1+/SOX9+/NKX6.1+ that clearly show that the C6 condition is preferable. Further elaboration of C6 could use such additional criteria. We comment on CDX2 / AFP in the discussion.

CDX2 should be quantified by IF / FACS.

We think that this would not add significantly since there are several other criteria, particularly the increase of the PDX1+/SOX9+/NKX6.1+ that clearly show that the C6 condition is preferable. Further elaboration of C6 could use such additional criteria. We comment on CDX2 / AFP in the discussion.

Karyotype analysis is good but not very precise when analyzing genetic micro alterations... what does a low-pass sequencing of the expanding lines look like? Are there any micro-deletions in the expanding lines?

This is an unusual request. Microdeletions may occur at any point – during passaging of hPS cells, differentiation as well as well as expansion but such data are so far not shown in publications – and reasonably so in our opinion. Thus, we have not done this analysis but it certainly would be appropriate in a clinical setting as part of QC.

Data supporting that the cells can be cryopreserved and recovered with >85% survival rate is not provided.

We now provide data for the C6-mediated expansion (Figure 4J). The freezing procedure was developed during the time we were testing C5 and we don’t have sufficient data to show reliably the survival of the cells during C5 expansion. Thus, we have now removed the reference in the C5 part of the manuscript.

Related to Figure 5:-Figure 5C - perhaps worth commenting on the different pathways that are enriched when cells undergo expansion and show some of the genes that are up/down regulated.

This is indeed of interest but since it will not address any specific question in the context of this work (eg is the endocrine program repressed?) and since it would not be followed by additional experiments we think that it would burden the manuscript unnecessarily. The data are accessible for any type of analysis through the GEO database.

Figure S5D shows in vitro clustering away from in vivo PP - it would be good to explain how in vitro generated PP differs from their in vivo counterparts instead of restricting the comparison to the in vitro protocol.

We have added a possible interpretation of this observation in the results section and discuss, how one could go properly about this comparison.

Quantification of Fig5F should be included. Is GP2 expression detectable by IF at p5 too?

We have quantified GP2 expression by FC at p10 but not at earlier stages. We include now the FC data in Fig5F

Validation of Fig5G by qPCR would be good. PDX1 did not seem reduced by IF in Figure 4.

The purpose of Fig5G is to compare the expression of the same genes across different expansion approaches. Therefore, in our view, qPCRs would not be appropriate since we do not have samples from the other approaches. We did not claim a reduction in PDX1 expression.

How can the authors explain the NGN3 expression at PP?

In our view, differentiation is a dynamic process and not all cells are synchronized at the same cell type, this is true in vivo and in vitro. Sc-RNA Seq data indeed show a small population of cells at PP that are NEUROG3+ (our unpublished data). We have now included this in the discussion.

Related to Figure 6:How do the different lines differ? Any statistical comparison between lines?

There is a paragraph dealing with the comparison of PP and ePP cells (p5 and p10) from different lines at the level of gene expression and the data are in Figure S6A-G. Then there is a paragraph addressing this at the level of PDX1/SOX9/NKX6.1 expression by FC. We have now expanded and rewrote the latter to include statistical comparisons across PPs from different lines at p0, p5 an p10

Related to Figure 7:Mention the use of micropatternedMicropatterned wells - not really correct. They use Aggrewells, micropatterned plates are something else.

We changed ‘micropatterned wells’ into ‘microwells’

Figure 7D, those are qPCR data. The label is inconsistent, why did they call it fold induction instead of fold change? Also, not sure if plotting the fold change to hPSC is the best here.

We use fold change when comparing the expression of the same gene at different passages but fold induction when comparing to its expression in hPS cells. We made sure it is also explained in the figure legends.

Absolute values should be shown for the GSIS to determine basal insulin secretion. Also, sequential stimulation to address if the cells are able to respond to multiple glucose stimulations.

We include now the secreted amounts of human C-peptide under the different conditions (Figure S7) normalized for cell numbers using their DNA content for the normalization. The many parameters we have used suggest that dPP and ePP SC-islets are very similar. If we were claiming a better S5-S7 procedure, such an assay would have been necessary but in this context, we think it is not absolutely necessary.

In vivo data would have strengthened the story. It is not clear if, in vivo, the cells will behave as the nonexpanded iPSC-derived beta cells.

We agree and these studies are under way but we do not expect to complete them soon. We feel that it is important that this work appears sooner rather than later.

**Reviewer #3 (Public Review)**
Summary:In this work, Jarc et al. describe a method to decouple the mechanisms supporting progenitor self-renewal and expansion from feed-forward mechanisms promoting their differentiation.The authors aimed at expanding pancreatic progenitor (PP) cells, strictly characterized asPDX1+/SOX9+/NKX6.1+ cells, for several rounds. This required finding the best cell culture conditions that allow sustaining PP cell proliferation along cell passages, while avoiding their further differentiation. They achieve this by comparing the transcriptome of PP cells that can be expanded for several passages against the transcriptome of unexpanded (just differentiated) PP cells.The optimized culture conditions enabled the selection of PDX1+/SOX9+/NKX6.1+ PP cells and their consistent, 2000-fold, expansion over ten passages and 40-45 days. Transcriptome analyses confirmed the stabilization of PP identity and the effective suppression of differentiation. These optimized culture conditions consisted of substituting the Vitamin A containing B27 supplement with a B27 formulation devoid of vitamin A (to avoid retinoic acid (RA) signaling from an autocrine feed-forward loop), substituting A38-01 with the ALK5II inhibitor (ALK5i II) that targets primarily ALK5, supplementation of medium with FGF18 (in addition to FGF2) and the canonical Wnt inhibitor IWR-1, and cell culture on vitronectin-N (VTN-N) as a substrate instead of Matrigel.Strengths:The strength of this work relies on a clever approach to identify cell culture modifications that allow expansion of PP cells (once differentiated) while maintaining, if not reinforcing, PP cell identity. Along the work, it is emphasized that PP cell identity is associated with the co-expression of PDX1, SOX9, and NKX6.1. The optimized protocol is unique (among the other datasets used in the comparison shown here) in inducing a strong upregulation of GP2, a unique marker of human fetal pancreas progenitors. Importantly GP2+ enriched hPS cell-derived PP cells are more efficiently differentiating into pancreatic endocrine cells (Aghazadeh et al., 2022; Ameri et al., 2017).The unlimited expansion of PP cells reported here would allow scaling-up the generation of beta cells, for the cell therapy of diabetes, by eliminating a source of variability derived from the number of differentiation procedures to be carried out when starting at the hPS cell stage each time. The approach presented here would allow the selection of the most optimally differentiated PP cell population for subsequent expansion and storage. Among other conditions optimized, the authors report a role for Vitamin A in activating retinoic acid signaling in an autocrine feed-forward loop, and the supplementation with FGF18 to reinforce FGF2 signaling.This is a relevant topic in the field of research, and some of the cell culture conditions reported here for PP expansion might have important implications in cell therapy approaches. Thus, the approach and results presented in this study could be of interest to researchers working in the field of in vitro pancreatic beta cell differentiation from hPSCs. Table S1 and Table S4 are clearly detailed and extremely instrumental to this aim.

We thank the reviewer for the positive assessment. Below we provide a point-by-point response to general comments and criticisms.

WeaknessesThe authors strictly define PP cells as PDX1+/SOX9+/NKX6.1+ cells, and this phenotype was convincingly characterized by immunofluorescence, RT-qPCR, and FACS analysis along the work. However, broadly defined PDX1+/SOX9+/NKX6.1+ could include pancreatic multipotent progenitor cells (MPC, defined as PDX1+/SOX9+/NKX6.1+/PTF1A+ cells) or pancreatic bipotent progenitors (BP, defined as PDX1+/SOX9+/NKX6.1+/PTF1A-) cells. It has been indeed reported that Nkx6.1/Nkx6.2 and Ptf1a function as antagonistic lineage determinants in MPC (Schaffer, A.E. et al. PLoS Genet 9, e1003274, 2013), and that the Nkx6/Ptf1a switch only operates during a critical competence window when progenitors are still multipotent and can be uncoupled from cell differentiation. It would be important to define whether culturing PDX1+/SOX9+/NKX6.1+ PP (as defined in this work) in the best conditions allowing cell expansion is reinforcing either an MPC or BP phenotype. Data from Figure S2A (last paragraph of page 7) suggests that PTF1A expression is decreased in C5 culture conditions, thus more homogeneously keeping BP cells in this media composition. However, on page 15, 2nd paragraph it is stated that "the strong upregulation of NKX6.2 in our procedure suggested that our ePP cells may have retracted to an earlier PP stage". Evaluating the co-expression of the previously selected markers with PTF1A (or CPA2), or the more homogeneous expression of novel BP markers described, such as DCDC2A (Scavuzzo et al. Nat Commun 9, 3356, 2018), in the different culture conditions assayed would more shield light into this relevant aspect.

This is certainly an interesting point. The RNA Seq data suggest that ePP cells resemble BP cells rather than MPCs and that this occurs during expansion. We have now added a new paragraph in the results section to illustrate this and added graphs of CPA2, PTF1A and DCDC2A expression during expansion in Figure 5, S5 as well as data in Table S5. In summary, we favor the interpretation that expanded cells are close but not identical to the BP identity and refer to that in the discussion. We have also amended the statement on page 15 stating

In line with the previous comment, it would be extremely insightful if the authors could characterize or at least discuss a potential role for YAP underlying the mechanistic effects observed after culturing PP in different media compositions. It is well known that the nuclear localization of the co-activator YAP broadly promotes cell proliferation, and it is a key regulator of organ growth during development. Importantly in this context, it has been reported that TEAD and YAP regulate the enhancer network of human embryonic pancreatic progenitors and disruption of this interaction arrests the growth of the embryonic pancreas (Cebola, I. et al. Nat Cell Biol 17, 615-26, 2015). More recently, it has also been shown that a cell-extrinsic and intrinsic mechanotransduction pathway mediated by YAP acts as gatekeeper in the fate decisions of BP in the developing pancreas, whereby nuclear YAP in BPs allows proliferation in an uncommitted fate, while YAP silencing induces EP commitment (Mamidi, A. et al. Nature 564, 114-118, 2018; Rosado-Olivieri et al. Nature Communications 10, 1464, 2019). This mechanism was further exploited recently to improve the in vitro pancreatic beta cell differentiation protocol (Hogrebe et al., Nature Protocols 16, 4109-4143, 2021; Hogrebe et al, Nature Biotechnology 38, 460-470, 2020). Thus, YAP in the context of the findings described in this work could be a key player underlying the proliferation vs differentiation decisions in PP.

We do refer to these publications now and refer to the YAP pathway in the introduction and results sections as well as in the discussion. We have not investigated more because the kinetics of the different components of the pathway are complex and do not give an indication of whether the pathway becomes more or less active – please see below.

**Author response image 2. sa3fig2:** 

Regarding the improvements made in the PP cell culture medium composition to allow expansion while avoiding differentiation, some of the claims should be better discussed and contextualized with current stateof-the-art differentiation protocols. As an example, the use of ALK5 II inhibitor (ALK5i II) has been reported to induce EP commitment from PP, while RA was used to induce PP commitment from the primitive gut tube cell stage in recently reported in vitro differentiation protocols (Hogrebe et al., Nature Protocols 16, 41094143, 2021; Rosado-Olivieri et al. Nature Communications 10, 1464, 2019). In this context, and to the authors' knowledge, is Vitamin A (triggering autocrine RA signaling) usually included in the basal media formulations used in other recently reported state-of-the-art protocols? If so, at which stages? Would it be advisable to remove it?

These points and our views are now included in the discussion

In this line also, the supplementation of cell culture media with the canonical Wnt inhibitor IWR-1 is used in this work to allow the expansion of PP while avoiding differentiation. A role for Wnt pathway inhibition during endocrine differentiation using IWR1 has been previously reported (Sharon et al. Cell Reports 27, 22812291.e5, 2019). In that work, Wnt inhibition in vitro causes an increase in the proportion of differentiated endocrine cells. It would be advisable to discuss these previous findings with the results presented in the current work. Could Wnt inhibition have different effects depending on the differential modulation of the other signaling pathways?

These points are now included in the discussion together with the points above

**Reviewer #3 (Recommendations For The Authors)**
Recommendations for improving the writing and presentation and minor comments on the text and figures:In the Introduction (page 3, line 1) it is stated: "Diabetes is a global epidemic affecting > 9% of the global population and its two main forms result from .....". The authors could rephrase/remove "global" repeated twice.

Corrected

On page 4 of the introduction, in the context of "Unlimited expansion of PP cells in vitro will require disentangling differentiation signals from proliferation/maintenance signals. Several pathways have been implicated in these processes..." the authors are advised to consider mentioning the YAP mediated mechanisms as another key aspect underlying MPC phenotype (Cebola, I. et al. Nat Cell Biol 17, 615-26, 2015) and the BP to endocrine progenitor (EP) commitment (Mamidi, A. et al. Nature 564, 114-118, 2018; Rosado-Olivieri et al. Nature Communications 10, 1464, 2019). This should be better discussed in the context of the Weaknesses mentioned in the Public Review. It would be worth considering adding effectors and other molecules involved in YAP and Hippo pathway signaling to Table S3.

We have added the role of the Hippo/YAP pathway in the introduction and mentioned in the results the finding that components of the pathway are generally not regulated except two that are now added in Table S3

In page 4, paragraph 3, near "and SB431542, another general (ALK4/5/7) TGFβ inhibitor", consider removing "another". SB431542 is the same inhibitor mentioned in the other protocols at the beginning of the paragraph.

The paragraph is rewritten because it was not clear – we used A83-01 and not SB431542. Other approaches had used SB431542.

Page 5, Table S2 is cited after Table S3, please consider reordering.

In fact, both S2 and S3 are relevant there, therefore we quote both now.

Page 8, 2nd paragraph, near "Expression of both AFP and CDX2 increased transiently upon expansion, at p5 (Figure S2H-J)." How do you explain results in FigS2C, D and FigS2E (AFP/CDX2)? RT-qPCR data does not suggest transient downregulation.

AFP and CDX2 were – wrongly – italicized in the quoted passage. Therefore, in one case we refer to the protein and in the other to the transcript levels. We corrected and added the qualifier ‘appeared’. The difference is most likely due to translational regulation but we did not elaborate since we do not know. In any case, we have used the, less favorable but more robust, gene expression levels as the main criterion.

Page 9, end of 2nd paragraph, Figure 5A is cited but it looks like this should be Figure 4A.

Corrected

Page 9, 3rd paragraph, when stating "C5 ePP cells of the same passage no..." please replace "no" with a number or a suitable abbreviation.

Corrected

Page 9, 3rd paragraph. Expressing the values in the Y axis in a consistent manner for FigS2B-D and FigS4A would make a comparison easier.

We strive to keep sections autonomous so that the reader would not have to flip between figures and sections – this is why we think that figure S4A is preferable as it is; it is a direct comparison of C6 to C5 for the different markers and has the additional advantage that one needs not to include p0 levels.

Page 9, 3rd paragraph. Green dots in FigS4A stand for p5 cells? if so, shouldn't these average 1 for all assayed genes?

No, because the baseline (average 1) is the C5 expression at the corresponding passage no. We changed the y-axis label, hopefully it is clearer now.

Page 10 3rd paragraph, please include color labels in Fig. 5G.

The different colors here correspond to the different expansion procedures that are compared. The samples are labelled on the x axis.

Page 10 3rd paragraph, Figure 6G is cited but it looks like this should be Figure 5G.

Corrected

Page 11, 1st paragraph, at "TF genes such as FOXA2 and RBJ remained comparable", please double check if "RBJ" should be "RBPJ".

Corrected

Page 11, end of 1st paragraph, when stating "Of note, expression of PTF1A was also undetectable in all ePP cells (Table S5)", is PTF1A expression level close to 1000 (which units?) in Table S5 considered undetectable?

This statement regarding ‘undetectable PTF1A expression’ refers to expanded PP cells (ePP), not PP cells at p0. For the latter, expression is indeed close to 1000 in normalized RNA-sequence counts as mentioned in the Table legend.

-Page 11, 4th paragraph, "In summary, the comparative transcriptome analyses suggested that our C6 expansion procedure is more efficient at strengthening the PP identity". In the context of comments made in the Public Review, more accuracy needs to be put when defining PP identity. Are these MPC or BP?

The RNA Seq data suggest that expansion promotes a MPC BP transition. We have added a paragraph in the corresponding results section and comment in the discussion.

Page 15, 2nd paragraph, the sentence "expression of PTF1A, recently shown to promote endocrine differentiation of hPS cells (Miguel-Escalada et al., 2022)" is confusing. Please double-check sentence syntax and reference. Does PTF1A expression "promote" or "create epigenetic competence" for endocrine differentiation?

Its role is in the MPCs and it prepares the epigenetic landscape to allow for duct and endocrine specification later, thus it ‘creates epigenetic competence’. The paper was cited out of context and we have now corrected it.

Additional recommendations by the Reviewing Editor:An insufficient number of experimental repetitions have been used for the following data: (Figure 1A, n = 2; Figures 2B-D, p10, n = 2; Figures 6A and B, VTN-N, n = 1).

This is true but we do not draw quantitative conclusions from or do comparisons with these data.